# Sleep-promoting neurons remodel their response properties to calibrate sleep drive with environmental demands

Stephane Dissel[1☉]*, Markus K. Klose[2☉], Bruno van Swinderen[3], Lijuan Cao[4], Melanie Ford[4], Erica M. Periandri[4], Joseph D. Jones[1], Zhaoyi Li[4], Paul J. Shaw[4]*

**1** Division of Biological and Biomedical Systems, School of Science and Engineering, University of Missouri-Kansas City, Kansas City, Missouri, United States of America, **2** University of Pittsburgh School of Medicine, Department of Pharmacology & Chemical Biology, Pittsburgh, Pennsylvania, United States of America, **3** Queensland Brain Institute, The University of Queensland, St Lucia, Australia, **4** Department of Neuroscience, Washington University School of Medicine, St. Louis, Missouri, United States of America

☉ These authors contributed equally to this work.
* dissels@umkc.edu (SD); shawp@wustl.edu (PJS)

**Data Availability Statement:** All relevant data are within the paper and its Supporting Information files.

## Abstract

Falling asleep at the wrong time can place an individual at risk of immediate physical harm. However, not sleeping degrades cognition and adaptive behavior. To understand how animals match sleep need with environmental demands, we used live-brain imaging to examine the physiological response properties of the dorsal fan-shaped body (dFB) following interventions that modify sleep (sleep deprivation, starvation, time-restricted feeding, memory consolidation) in *Drosophila*. We report that dFB neurons change their physiological response-properties to dopamine (DA) and allatostatin-A (AstA) in response to different types of waking. That is, dFB neurons are not simply passive components of a hard-wired circuit. Rather, the dFB neurons intrinsically regulate their response to the activity from upstream circuits. Finally, we show that the dFB appears to contain a memory trace of prior exposure to metabolic challenges induced by starvation or time-restricted feeding. Together, these data highlight that the sleep homeostat is plastic and suggests an underlying mechanism.

## Introduction

The importance of sleep is highlighted by the observation that it is evolutionarily conserved despite directly competing with all motivated waking behaviors [1,2]. Not only does sleep compete with foraging, eating, and mating [3–6], for example, high sleep drive may be maladaptive in many circumstances since falling asleep could place the individual in danger of immediate physical harm [7,8]. On the other hand, sleep plays a critical role in learning and memory, supports adaptive behavior, and facilitates creative insight [9–12]. Together, these observations suggest that it will not be possible to fully understand sleep's function without knowing how sleep circuits calibrate sleep drive with motivational states.

**Funding:** This work was funded by the National Institute of Neurological Disorders and Stroke 5R01NS051305-14 and 076980-08 to PJS. The funders had no role in study design, data collection and analysis, decision to publish, or preparation of the manuscript.

**Competing interests:** The authors have declared that no competing interests exist.

**Abbreviations:** AstA, allatostatin-A; cAMP, cyclic adenosine monophosphate; DA, dopamine; dFB, dorsal fan-shaped body; PDF, pigment-dispersing factor; RNAi, RNA interference; TTX, tetrodotoxin; vFB, ventral fan-shaped body.

In recent years, a great deal of progress has been made dissecting circuits that regulate motivated behavior in flies [13–17]. Typically, the properties of a circuit are examined as a function of one internal state. For example, sleep circuits are evaluated after sleep loss [18–20], feeding in response to starvation or high dietary sugar [21,22], thirst following water deprivation [23], mating after social isolation [24], etc. However, since internal states can promote conflicting goal-directed behaviors, recent studies have begun to evaluate how circuits regulate competing activities such as feeding and sleep [5,25–29], thirst and hunger [23], sweet and bitter taste [30], hunger and mating [31], and thirst versus hunger relevant memory [23], for example.

A common theme that has emerged from these studies has been that a specific deprivation-state differentially activates a subset of peptidergic neurons that then modulate classic neurotransmitter systems, frequently dopamine (DA), to alter downstream circuits and thus motivated behavior [31–33]. For example, water deprivation preferentially activates a subset of peptidergic neurons that inhibit specific dopaminergic neurons to alter thirst-relevant memory [34]. In this model, the competition between state-specific peptidergic neurons will determine which motivated behavior will be expressed. An open question, however, is whether the neurons receiving peptidergic or dopaminergic input are constrained such that their responses are determined by upstream signals, or on the other hand, whether they can change their own response properties over time to influence a given outcome.

Sleep-promoting *R23E10* neurons are well suited to investigate these relationships. *R23E10* neurons are modulated both by dopaminergic neurons and allatostatin-A (AstA)-expressing circadian-neurons [19,23,25–27,35,36]. AstA-expressing neurons promote sleep by releasing glutamate onto *R23E10* neurons [25]. AstA is functionally similar to galanin and has been implicated in both feeding and sleep [27,37]. Sleep-promoting *R23E10* neurons comprise a sleep switch and are an integral part of the sleep homeostat [19,26]. The intrinsic properties of *R23E10* neurons have been evaluated in the context of sleep loss where it seems they monitor redox processes as an indicator of energy metabolism [19,38,39]. Thus, *R23E10* neurons may serve as command neurons of sorts that can integrate stimuli from competing internal states to gate information flow [40,41]. Both the input and the output connections to *R23E10* are well documented [25,26,35,36]. Thus, the focus of this study is to evaluate sleep-promoting *R23E10* neurons following challenges that modulate motivational states.

In this study, we identify independent sets of heterogeneous sleep-promoting neurons that change their responses to DA and AstA following challenges that influence sleep. In addition, we identify a wake-promoting effect of AstA on *R23E10* neurons suggesting that the co-release of inhibitory AstA with excitatory glutamate may attenuate the overexcitement of *R23E10* neurons during high sleep drive and allow animals to maintain wakefulness in dangerous or life-threatening conditions. Finally, we find that both time-restricted feeding and acute starvation enhance subsequent waking by remodeling the expression of DA receptors in sleep-promoting neurons. Together, these data provide new insights into the interaction between internal states and sleep regulation.

## Results

### Neurons projecting to the dorsal fan-shaped body are modulated by allatostatin

Fan-shaped body (dFB)-projecting *R23E10* neurons are an important component of the sleep homeostat [19,38,39]. Sleep-promoting dFB neurons are believed to be inhibited by wake-promoting dopaminergic neurons and activated when glutamate is released from sleep-promoting AstA-expressing neurons [25,35,36]. Surprisingly, the role of AstA on *R23E10* neurons has not been investigated. Thus, we used behavioral genetics and live-brain imaging to characterize

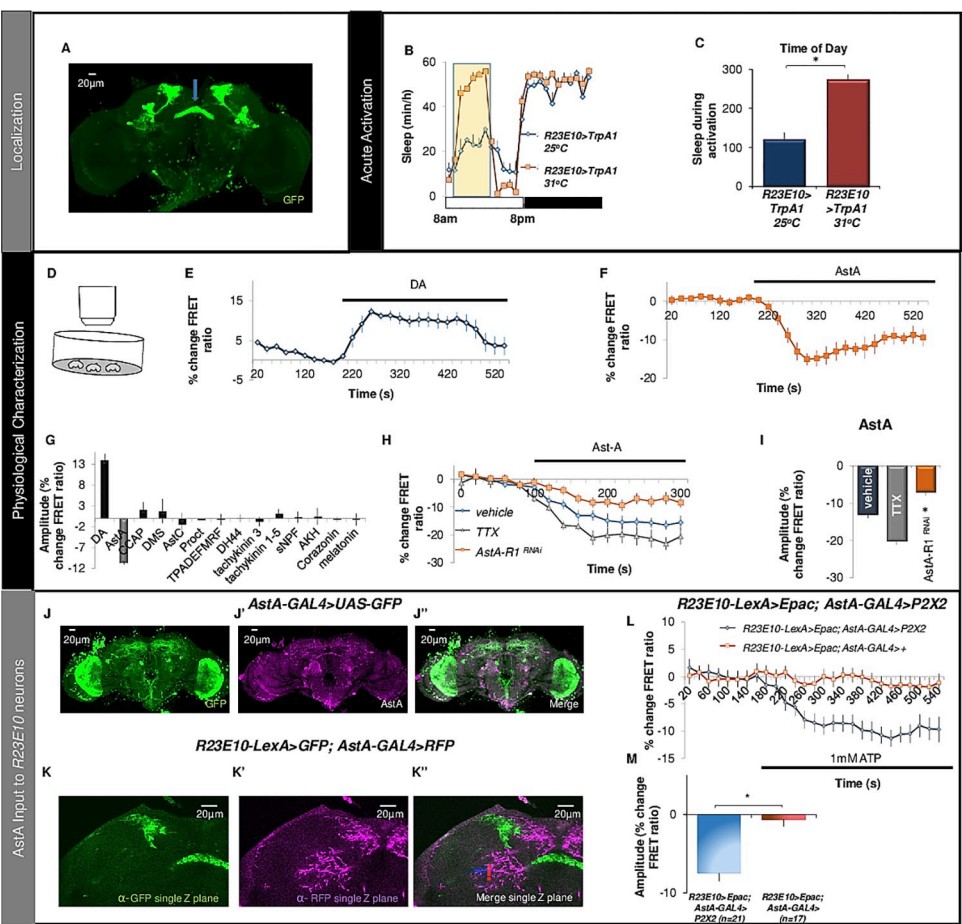

**Fig 1. Sleep-promoting *R23E10* neurons respond to AstA.** (**A**) Confirmation of *R23E10-GAL4>UAS-GFP* dFB projections (blue arrow). (**B, C**) *R23E10-GAL4>UAS-dTrpA1* showed a significant increase in sleep at 31°C (red) compared to siblings maintained at 25°C (blue; *n* = 16/group). (**D**) Schematic of high-throughput Epac imaging. (**E, F**) Normalized FRET ratio in *R23E10-GAL4>UAS-Epac1-camps* in response to DA ($3 \times 10^{-5}$ M) and AstA ($1e^{-6}$ M) (*n* = 3–7 cells/condition); (**G**) % change in FRET ratio in *R23E10-GAL4>UAS-Epac1-camps* following the application of neuroactive compounds (*n* = 6–21 cells/condition). (**H**) TTX ($1e^{-7}$ M) does not prevent the response of *R23E10>UAS-Epac1-camps* to bath applied AstA ($1e^{-6}$ M) (*n* = 3–6 cells/condition). Knocking down *AstA-R1* attenuates the response of *R23E10* neurons to AstA (red trace; *n* = 6). (**I**) Quantification of H (ANOVA $F_{[2,16]}$ = 15.4, *p* = 0.000183784; *p* < 0.05, modified Bonferroni test). (**J**) Confocal stack of an *AstA-GAL4>UAS-GFP* fly brain stained with anti-GFP (**J'**) Anti-AstA (magenta) antibody. (**J''**) A merge image. (**K**) Single optical section (0.5 μm) of a *R23E10-LexA>LexAop-GFP, AstA-GAL4>UAS-RFP* fly brain stained with anti-GFP (**K'**) Anti-RFP (magenta). (**K''**) A merge image. (**L**) Application of 1 mM ATP induced a decrease in cAMP levels in *R23E10* cells when the P2X2 receptor was expressed in *AstA-GAL4*-expressing neurons (blue trace), but not in control lines lacking the P2X2 receptor (red trace). (**M**) Quantification of L. Error bars represent standard error of the mean (SEM). Underlying data is in S1 Data and S1 Datasheet. AstA, allatostatin-A; cAMP, cyclic adenosine monophosphate; DA, dopamine; dFB, dorsal fan-shaped body; TTX, tetrodotoxin.

the effects of AstA and DA on *R23E10* neurons. The expression pattern of *R23E10* neurons is shown in Fig 1A. Expressing the temperature-sensitive *transient receptor potential cation channel* (*UAS-TrpA1*) in *R23E10* neurons and raising the temperature from 25°C to 31°C for 6 h increased sleep (Fig 1B and 1C). The parental controls do not show an increase in sleep during the exposure to 31°C (S1A Fig). Since many neuromodulators act through second messenger signaling cascades, we expressed the cyclic adenosine monophosphate (cAMP) sensor, *UAS-Epac1-camps*, in *R23E10* neurons and used high-throughput live-brain imaging to monitor neuronal responses to bath applied DA (Fig 1D) [42,43]. As seen in Fig 1E, DA increases

cAMP levels in dFB neurons as previously reported [36]. It is important to note that while *R23E10>GFP* identifies approximately 14 neurons/hemisphere, *R23E10>UAS-Epac1-camps* reliably labeled approximately 8 neurons/hemisphere (S1B Fig). Thus, we have confirmed that *R23E10* neurons are sleep-promoting and respond to DA.

Given that *R23E10* neurons are downstream of AstA-expressing neurons [25], we evaluated the response properties of *R23E10* neurons to AstA and 14 other compounds that can influence motivational states [44]. As seen in Fig 1F and 1G, AstA reduced cAMP levels in *R23E10* neurons. Previous studies indicate that AstA is an inhibitory peptide [45]. Individual traces for Fig 1G are plotted in S2A Fig. Consistent with the cAMP responses, knocking down a battery of neuropeptide receptors in *R23E10* neurons using RNA interference (RNAi) did not substantially modify sleep (S2B Fig). To determine whether the effects of AstA on *R23E10* neurons are direct, we incubated brains in tetrodotoxin (TTX), which inhibits the firing of action potentials. TTX does not prevent the AstA-mediated cAMP response of *R23E10* cells (Fig 1H, green trace). To further confirm a direct role of AstA, we used RNAi to knock down the *AstA-R1* receptor in *R23E10* neurons. As seen in Fig 1H (red trace), the cAMP response triggered by application of AstA is attenuated. Quantification of these effects is shown in Fig 1I. Importantly, *R23E10* neurons responded normally to DA when *AstA-R1* levels are knocked-down indicating that the reduced response to AstA is not the result of a nonfunctional neuron (S2C Fig).

To better understand the relationship between AstA and *R23E10* neurons, we used immunohistochemistry to evaluate the overlap between AstA and *AstA-GAL4*. As previously reported, *AstA-GAL4* does not fully recapitulate the AstA expression pattern within the projections to the dFB (Fig 1J–1J") [37]. Nonetheless, *AstA-GAL4* is expressed near the dendritic fields of *R23E10* suggesting a physical connection between the 2 group of neurons (Fig 1K–1K") [25]. To evaluate functional connectivity of the *AstA-GAL4*, *R23E10* circuit, we expressed the P2X2 activator [46] in *AstA-GAL4* neurons while measuring cAMP with *LexAop-Epac* in *R23E10-LexA* neurons. As seen in Fig 1L and 1M, perfusion of ATP that activates *P2X2*, leads to a reduction of cAMP levels in *R23E10* cells, mimicking the effect of AstA on *R23E10* neurons reported above (Fig 1F). No changes in cAMP signaling were observed in parental controls perfused with ATP but that do not express *P2X2* indicating the effects are specific to the activation of *AstA-GAL4* neurons (Fig 1L, red trace). Thus, we show that the cAMP response of *R23E10* neurons to AstA is similar to that seen when AstA-expressing cells are activated.

The sleep-promoting effects of AstA-expressing neurons have been shown to be due to the release of glutamate onto *R23E10* neurons [25]. Because AstA is an inhibitory neuropeptide [44,45], we would predict the impact of AstA on *R23E10* neurons would be wake promoting and that knocking down *AstA-Rs* would thus increase sleep. To test this hypothesis, we evaluated sleep after knocking down *AstA-R1* or *AstA-R2* in *R23E10* neurons. As seen in Fig 2A and 2B and 2E and 2F, total sleep is increased when *AstA-R1* or *AstA-R2* is knocked down in *R23E10* neurons, suggesting that the dFB is under consistent allatostatinergic inhibitory tone. Importantly, sleep is also more consolidated during the day (Fig 2C and 2G); sleep consolidation is also increased at night (S3A and S3B Fig). Furthermore, the latency to fall asleep at night is reduced (S3C and S3D Fig). The increase in sleep is not due to unhealthy or sick flies since waking activity is not reduced compared to parental controls (Fig 2D and 2H). To rule out off-target effects of the single RNAi line, we tested 3 additional, independent *AstA-R1* RNAi lines and found that sleep is significantly increased in each line (S3E Fig). Given that increasing the activity of AstA neurons has been reported to increase sleep, but the *R23E10>AstA-R*[RNAi] data indicate that AstA provides a wake signal, we asked whether our results might be due to unknown environmental factors in our laboratory. To evaluate this possibility, we utilized 2 lines (*AstA-GAL4* and *R65D05-LexA*) that express in AstA-expressing

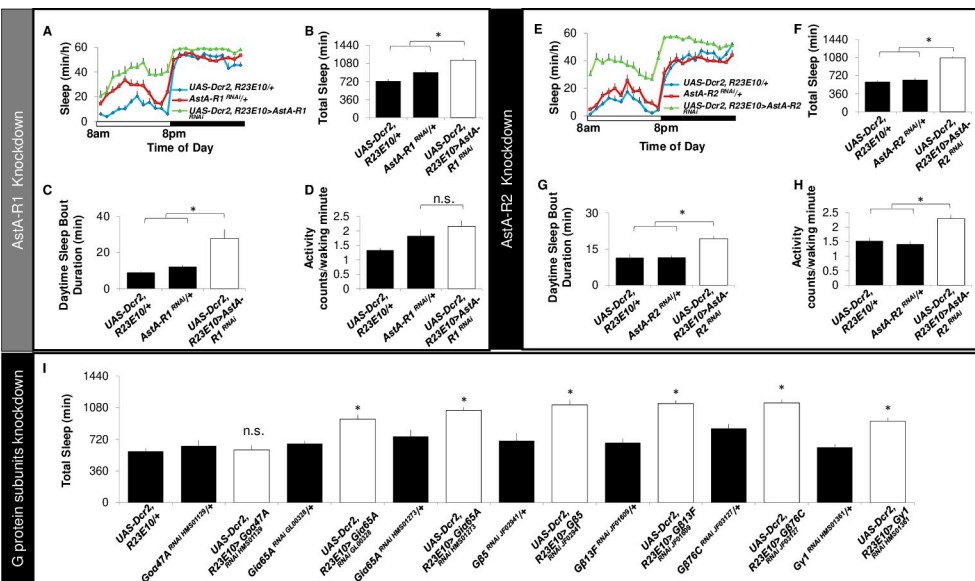

**Fig 2. Knocking down *AstA-R1* or *AstA-R2* in *R23E10* neurons increases sleep.** (**A**) Sleep in minutes per hour, (**B**) total sleep, and (**C**) daytime sleep-bout duration are increased when *AstA-R1* is knocked down in *R23E10* neurons compared with both *UAS-Dcr2, R23E10-GAL4/+*, and *AstA-R1^RNAi^/+* parental controls (*n* = 16/condition, $^*p < 0.05$, modified Bonferroni test). (**D**) Waking activity is not different between *UAS-Dcr2, R23E10-GAL4/+> AstA-R1^RNAi^/+* experimental flies, and both *UAS-Dcr2, R23E10-GAL4/+*, and *AstA-R1^RNAi^/+* controls (*n* = 16/condition, *p* > 0.05, modified Bonferroni test). (**E**) Sleep in minutes per hour, (**F**) total sleep, and (**G**) daytime sleep bout duration are increased when *AstA-R2* is knocked down in *R23E10* neurons compared with both *UAS-Dcr2, R23E10-GAL4/+*, and *AstA-R2^RNAi^/+* parental controls (*n* = 16/condition, $^*p < 0.05$, modified Bonferroni test). (**H**) Waking activity is significantly increased in *UAS-Dcr2, R23E10-GAL4/+> AstA-R2^RNAi^/+* experimental flies compared to both *UAS-Dcr2, R23E10-GAL4/+*, and *AstA-R2^RNAi^/+* controls (*n* = 16/condition, *p* < 0.05, modified Bonferroni test). (**I**) Total sleep is not changed when Goα47A levels are reduced in *R23E10* neurons, but independently knocking down Giα65A, Gβ5, Gβ13F, Gβ76C, and Gγ1 in *R23E10* cells increase total sleep compared with parental controls (*n* = 16/condition; $^*p < 0.05$). Error bars represent SEM. Underlying data is in S2 Data and S1 Datasheet.

clock neurons and increase sleep [25]. As seen in S3F Fig, both *AstA-GAL4-dTrpA1* and *R65D05-LexA>LexAop-dTrpA1* lines substantially increased sleep consistent with previous reports [25]. Because AstA is an inhibitory neuropeptide, and similar mammalian receptors signals mainly through the Gi pathway [47], we hypothesized that the AstA receptors might be coupled to inhibitory G protein subunits. To test this hypothesis, we used RNAi to knockdown specific G protein subunits in sleep-promoting dFB neurons. As seen in Fig 2I, knocking down *Goα47A* had no effect on sleep, while knocking down *Giα65A* using 2 different RNAi lines significantly increased sleep. Furthermore, independently knocking down β and γ1 subunits in sleep-promoting *R23E10* neurons also increased sleep (Fig 2I). Thus, these data suggest that the inhibition of dFB neurons can be achieved via coupling with inhibitory G proteins.

## Sleep-promoting dFB neurons are diverse

dFB neurons have been hypothesized to gate different aspects of sleep such as locomotion, sensory thresholds, etc. [26,48]. This hypothesis suggests that the dFB could be comprised of independent sets of sleep-promoting neurons that each respond to distinct environmental challenges. If this were to be the case, then we should be able to identify novel dFB projecting GAL4 lines. We obtained 12 *GAL4* lines from the Flylight collection that were selected to match the expression pattern of the original dFB sleep-promoting GAL4 lines (*104y* and *C5*)

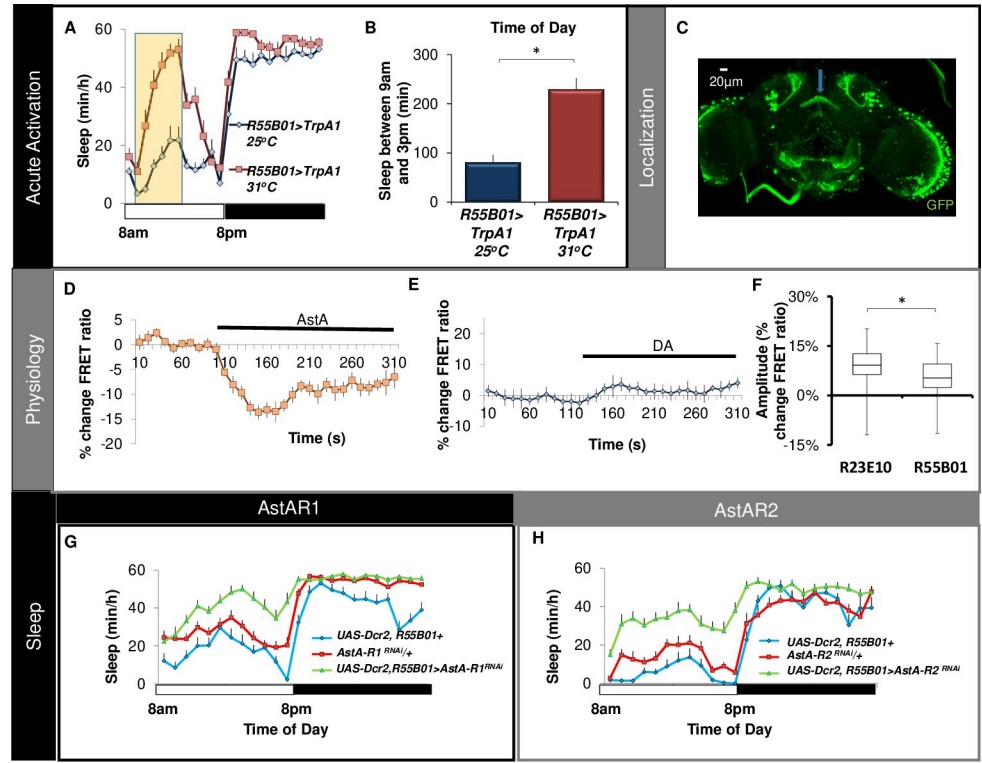

**Fig 3. *R55B01* neurons respond to AstA.** (**A, B**) *R55B01-GAL4>UAS-dTrpA1* flies showed a significant increase in sleep at 31˚C (red) compared to siblings maintained at 25˚C (blue; *n* = 16/group). (**C**) Expression pattern of *R55B01-GAL4>UAS-GFP* (blue arrow indicates the location of dFB). (**D**) Normalized FRET ratio in *R55B01>UAS-Epac1-camps* in response to $1e^{-6}$ M AstA (*n* = 23 cells). (**E**) Normalized FRET ratio in *R55B01>UAS-Epac1-camps* to $3e^{-5}$ M DA (*n* = 10). (**F**) Box plot comparing the response properties of *R23E10> UAS-Epac1-camps* and *R55B01> UAS-Epac1-camps* neurons to $3e^{-5}$ M DA. The bottom and top of each box represents the first and third quartile, and the horizontal line dividing the box represents the median. The whiskers represent the minimum and maximum individual cell responses; *n* = 68 cells for *R23E10* and *n* = 115 cells for *R55B01*. (**G, H**) Knocking down *AstA-R1* or *AstA-R2* in *R55B01* neurons increases sleep. Sleep in minutes per hour in *R55B01-GAL4>AstAR1*<sup>RNAi</sup> and *R55B01-GAL4>AstAR2*<sup>RNAi</sup> (*n* = 16/condition). Error bars represent SEM. Underlying data is in S3 Data and S1 Datasheet. AstA, allatostatin-A; DA, dopamine; dFB, dorsal fan-shaped body.

[10,49]. We activated these neurons by expressing *UAS-TrpA1* as described above. Surprisingly, only 1 driver, *R55B01* increased sleep similarly to *R23E10* when compared to siblings maintained at 25˚C (Figs 3A and 3B and S4A). The parental controls do not show an increase in sleep during the 6-h exposure to 31˚C (S1 Fig). Similar results were obtained when expressing the sodium bacterial channel, *UAS-NaChBac* (S4B Fig). As seen in Fig 3C, *R55B01* project to the dFB neuropil (blue arrow) and have cell bodies located in the same anatomical region in the brain as *R23E10* neurons. To determine potential overlap between *R23E10* and *R55B01*, we conducted co-labeling experiments by expressing RFP with GAL4/UAS and GFP with LexA/LexAop in the same fly. As seen in S5C–S5C" and S5D Fig *R23E10-LexA* is similar, but not identical, to the expression pattern of *R23E10-GAL4*. Importantly, *R23E10-LexA* and *R55B01*-GAL4 only share approximately 4 neurons in common (S5A–S5A" and S5B Fig). As with *R23E10* neurons, *R55B01>UAS-Epac1-camps* labels fewer dFB projecting neurons (approximately 8 neurons) compared to *R55B01>GFP*. Similar to *R23E10* neurons, *R55B01>UAS-Epac1-camps* show robust responses to AstA (Fig 3D). Moreover, AstA positive staining is co-localized with the GFP positive processes of *R55B01* neurons (S5E–S5E" Fig). In

contrast to *R23E10* neurons, the response of *R55B01* neurons to DA is either reduced or absent (Fig 3E). A box plot comparing the range of responses of *R23E10* and *R55B01* neurons to DA is shown in Fig 3F. Given the robust responses of *R55B01* neurons to AstA, we evaluated sleep after knocking down *AstA-R1* or *AstA-R2*. As seen in Fig 3G and 3H, sleep is increased upon knocking down either *AstA-R1* or *AstA-R2* in *R55B01* neurons. Quantification of sleep architecture in *R55B01>AstAR1^{RNAi}* and *R55B01>AstAR2^{RNAi}* can be found in S6 Fig. Thus, we have identified an additional sleep-promoting driver that projects to the dFB, has little overlap with *R23E10* neurons, and is less responsive to DA.

### *R23E10* neurons differentially integrate sleep-relevant stimuli

Together with the literature, these data identify 2 major wake-promoting signals that impact *R23E10* neurons, AstA and DA [35,36]. Although both signals are inhibitory, they derive from neuronal circuits with opposite functions; AstA is released from sleep-promoting neurons while DA is released from wake-promoting neurons [25,35,36]. Given these divergent roles, we hypothesize that AstA and DA will operate largely independently. To test this hypothesis, we used the live-brain imaging approach described above (Fig 1D) and evaluated the response of *R23E10* neurons to either DA or allatostatin during conditions that alter sleep drive. Because *R55B01* neurons respond less to DA, we focused solely on *R23E10* neurons; *R55B01* neurons will be studied later. We first evaluated the response properties of *R23E10* neurons in unperturbed flies under conditions characterized by large changes in sleep time (e.g., ontogeny, gender, individual differences). We then evaluated the response properties of *R23E10* neurons after experimental interventions that modulate sleep drive (e.g., sleep loss, starvation, training that induces long-term memory) [3,28,50–53] (Fig 4).

As with humans, sleep is highest in young flies and then stabilizes in early adulthood. Interestingly, *R23E10* neurons are less responsive to DA in 0- to 1-day-old flies compared to 6- to 8-day-old mature adults; no changes were observed in response to AstA (Fig 4A and 4B). The response properties of *R23E10* neurons to DA were similar in 6- to 8-day-old flies and 30- to 38-day-old flies S7 Fig. In contrast to age, we did not observe any changes in the response properties of *R23E10* neurons in male and female flies despite large sexual dimorphisms in sleep behavior [52,54]. We have previously shown that individual differences in sleep time can be exploited to evaluate sleep regulation and function [55]. Importantly, our previously published data indicate that spontaneously short-sleeping flies appear to experience both high wake-drive and high sleep drive simultaneously [56]. With that in mind, we evaluated the response properties of *R23E10* neurons in spontaneously short-sleeping *R23E10>UAS-Epac1-camps* (i.e., total sleep time less than 400 min) compared to normal sleeping siblings (800 to 960 min sleep). As seen in Fig 4C, *R23E10* neurons appear to be under stronger inhibitory tone from AstA in spontaneous short sleepers compared to normal sleeping siblings. Although one would predict a stronger response of *R23E10* neurons to the wake-promoting effects of DA, short-sleeping flies were less responsive to DA (Fig 4D). Future studies will be needed to determine whether incongruent responses of *R23E10* neurons to AstA and DA reveal underlying deficits in sleep regulation.

We next asked whether distinct experimental interventions that modulate sleep drive would alter the physiological response properties of *R23E10* neurons in similar or dissimilar ways. Both sleep deprivation and extended periods of starvation are followed by a compensatory increase in sleep [3,28,57]. To evaluate how sleep disruption would influence *R23E10* neurons, flies were sleep deprived for 12 h or starved for 18 h and compared to untreated siblings. As seen in Fig 4E, cAMP responses to AstA are increased following sleep deprivation but were unchanged in response to DA (Fig 4F). Surprisingly, the response of *R23E10* neurons to either

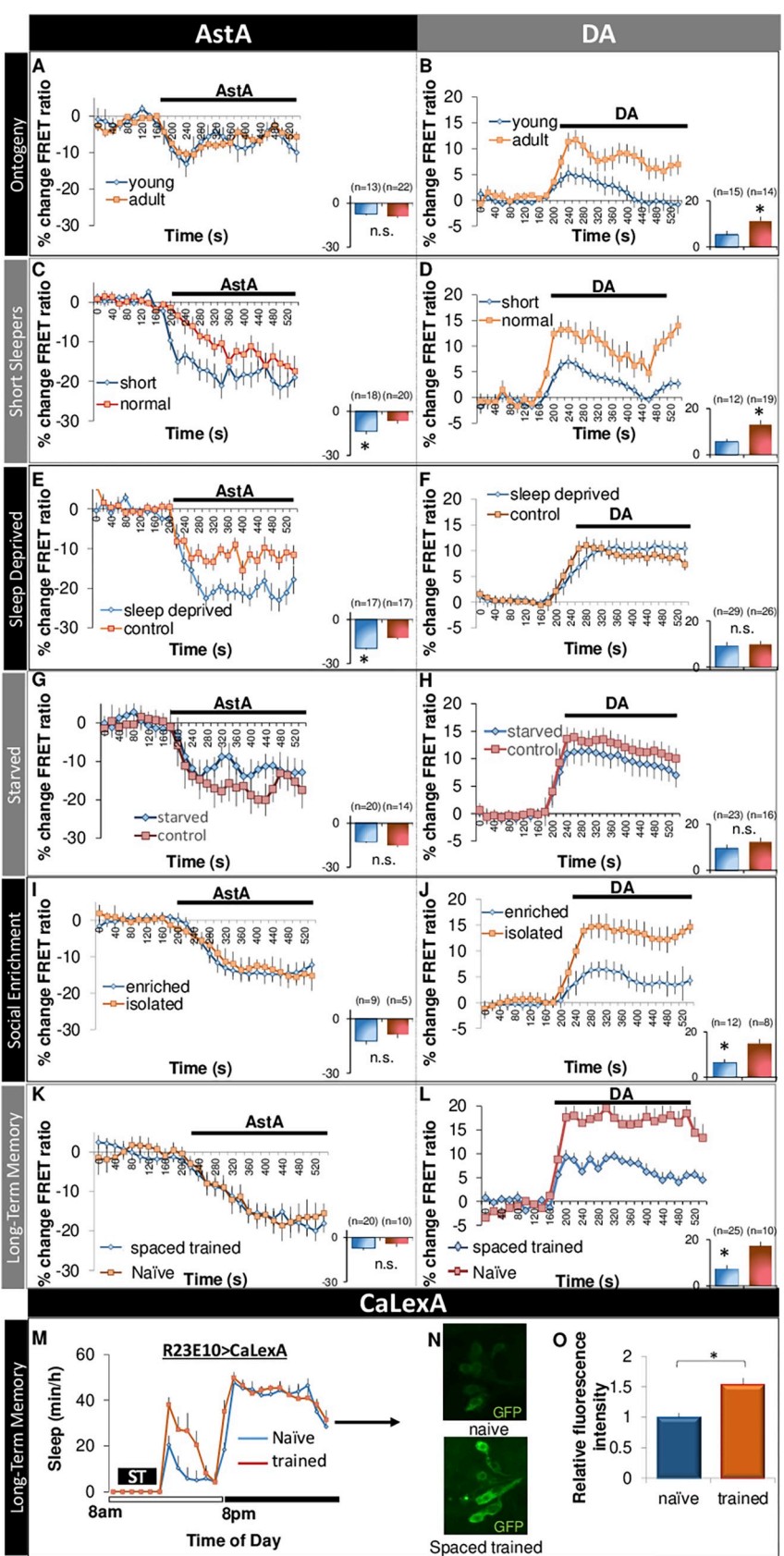

**Fig 4. Sleep-promoting neurons integrate a variety of relevant stimuli.** (**A, B**) Response of *R23E10* neurons in young (0–1 day old) and adult (6–8 day old) flies in response to AstA ($10^{-6}$ M) ($n = 13, 22$ cells) or DA ($3e^{-5}$ M) ($n = 15, 14$ cells). (**C, D**) Response of *R23E10* neurons in spontaneously short-sleeping flies (<400 min/day) and normal sleeping siblings (600–900 min/day) (AstA, $n = 18, 20$ cells) and (DA, $n = 12, 19$ cells). (**E, F**) Response of *R23E10* neurons following 12-h sleep deprivation compared to untreated siblings (AstA, $n = 17, 17$ cells) and (DA, $n = 29, 26$ cells). (**G, H**) Response of *R23E10* neurons following 18 h starvation compared to untreated siblings; (AstA, $n = 20, 14$ cells) and (DA, $n = 23, 15$ cells). (**I, J**) Response of *R23E10* neurons following social enrichment compared to isolated siblings; (AstA, $n = 9, 5$ cells) and (DA, $n = 12, 8$ cells). (**K, L**) Response of *R23E10* neurons following spaced training compared to naïve controls (AstA, $n = 4, 7$ cells) and (DA, $n = 25, 10$ cells). (**M**) Sleep is increased in *R23E10-GAL4>CaLexA* flies following a spaced courtship training ($n = 10$ per group, $p < 0.05$). (**N**) Representative confocal stack of the brains of a naïve (top) and spaced trained (bottom) *R23E10-GAL4>CaLexA* flies stained with anti-GFP antibody. Flies were dissected on the day after courtship conditioning. (**O**) Quantification of GFP staining intensity in R23E10 neurons in naïve and spaced trained *R23E10-GAL4>CaLexA* flies ($n = 107$ neurons for naïve and 100 neurons for trained). Insets are quantification of the corresponding traces. Underlying data is in S4 Data and S1 Datasheet. AstA, allatostatin-A; DA, dopamine.

AstA or DA were not changed while being starved (Fig 4G and 4H and see below). In contrast to sleep deprivation and starvation, social enrichment induces plasticity in specific neural circuits to increase sleep without exposing flies to sleep loss [51,58]. Changes in sleep following social enrichment have been mapped to pigment-dispersing factor (PDF)-expressing clock neurons [58]. Interestingly, AstA signaling is modulated by PDF [27]. Thus, to evaluate the impact of social rearing, we housed flies in a socially enriched environment (50 flies/vial) for 5 days and evaluated cAMP responses compared to isolated siblings. As seen in Fig 4I, *R23E10* neurons of socially enriched or isolated flies show similar responses to AstA. However, DA responses of *R23E10* cells are strongly reduced by social enrichment (Fig 4J). These data demonstrate that disparate behavioral manipulations that increase sleep drive in different ways induce independent physiological responses of *R23E10* neurons to AstA and DA.

Sleep-dependent memory consolidation has been linked to both the ventral and the dFB [10,59]. Thus, we asked whether *R23E10* neurons would modify their physiological responses to a training protocol that induces LTM [9]. As seen in Fig 4K, responses of *R23E10* neurons to AstA are not different following courtship conditioning. However, training resulted in a dramatic reduction in the response of *R23E10* neurons to DA (Fig 4L). Interestingly, the effect of training on DA responses could still be observed 24 h after the end of the training only returning to baseline 48 h later suggesting long-term plastic changes in *R23E10* neurons (S8 Fig). To determine whether the changes in cAMP were due to nonspecific effects of courtship or to memory consolidation, we evaluated cAMP levels following a massed training protocol consisting of a single 3-h session that does not result in LTM formation [10]. As seen in S8 Fig, massed training did not alter the responses of *R23E10* cells to DA. Furthermore, no changes in responses to DA were found when training was followed by 4 h of sleep deprivation S8 Fig. To further evaluate the role of the dFB in courtship memory, we used *CaLexA* (*calcium-dependent nuclear import of LexA*) to see if *R23E10* neurons might show sustained activity following a spaced training protocol [60]. *R23E10* flies expressing *CaLexA* were exposed to a training protocol consisting of $3 \times 1$ h individual pairings of a naïve male with a non-receptive female target separated by a rest period of 1 h. As seen in Fig 4M, sleep is increased following training compared with non-trained siblings consistent with previous reports [51,58]. *R23E10* neurons show higher GFP signal in trained animals when assessed the following morning compared with their naïve counterparts indicating that sleep-promoting neurons are more active following courtship memory training (Fig 4N and 4O for quantification). Together, these data indicate that the response properties of *R23E10* neurons display long-lasting changes following protocols that induce LTM.

## Prior feeding experience alters the recruitment of DA receptors to modulate sleep

Initial studies indicated that the wake-promoting effects of DA on dFB neurons are mediated by *Dopamine 1-like receptor 1* (*Dop1R1*) [35,36,61]. However, the role of *Dop1R1* has been called into question [19]. We have recently shown that the constellation of receptors expressed on a neuron can be altered by starvation to include a receptor that is not typically present [3]. Specifically, our data suggest that the new receptor is recruited to amplify wake-promoting signals in clock neurons to allow animals to engage in adaptive waking behaviors. To determine whether the phenomenon of recruiting new wake-promoting receptors to a neuron will generalize to the dFB, we re-examined the role of *Dop1R1* in *R23E10* neurons. In addition, we also evaluated *R55B01* neurons because these cells have a limited response to DA. We hypothesized that if *Dop1R1* is needed to support waking following a metabolic challenge, knocking it down would manifest as an increase in sleep. Since 18 h of starvation did not change the response properties of *R23E10* neurons (Fig 4G and 4H), we used time-restricted feeding to safely impose an alternate challenge of longer duration [62–64]. The time-restricted feeding protocol is shown in Fig 5A. Flies are only given access to food between 8 AM and 5 PM for a total of 7 days (restricted). After time-restricted feeding, flies are placed into Trikinetics tubes where they were allowed to eat ad lib while sleep is evaluated. Siblings that were maintained in vials with standard food available ad lib and flipped at the same times as their restricted counterparts served as treatment controls (Fig 5A). Consistent with previous reports [19,36], knockdown of *Dop1R1* in *R23E10* neurons did not alter sleep in flies that were able to feed ad lib (Fig 5B and 5D and 5E). Similarly, no changes in sleep were observed in untreated *R55B01>Dop1R1^{RNAi}* flies (Fig 5C–5E). However, both *R23E10>Dop1R1^{RNAi}* and *R55B01>Dop1R1^{RNAi}* flies displayed dramatic increases in sleep following 7 days of time-restricted feeding compared to their *R23E10/+*, *R55B01/+*, and *Dop1R1^{RNAi}/+* parental controls (Fig 5F–5I). Changes in sleep during time-restricted feeding are shown in S9 Fig. These data are consistent with the hypothesis that, under certain circumstances, a new receptor can be recruited to amplify wake-promoting signals.

Time-restricted feeding is characterized by defined intervals of feeding and fasting. Thus, during time-restricted feeding, *Dop1R1* may be recruited to *R23E10* neurons to either support waking during starvation or to support waking when food is available. To distinguish between these possibilities, we monitored sleep in *R23E10>Dop1R1^{RNAi}* and *R55B01>Dop1R1^{RNAi}* flies and their parental controls during baseline, during 18 h of starvation, and for 3 days after being placed back onto food (Fig 6A and 6B). As above, no changes in baseline sleep were observed while knocking down *Dop1R1* in either *R23E10* or *R55B01* neurons compared to parental controls (Fig 6A–6D). Moreover, *R23E10>Dop1R1^{RNAi}* and *R55B01>Dop1R1^{RNAi}* flies exhibited similar sleep patterns to parental controls during starvation (S10A Fig). However, during recovery following 18 h of starvation, an increase in sleep was observed in both *R23E10>Dop1R1^{RNAi}* and *R55B01>Dop1R1^{RNAi}* flies compared to their respective parental controls (Fig 6A and 6B). It is important to highlight that sleep in the experimental lines only diverged from their parental controls after several hours, or more, of recovery (Fig 6A and 6B arrows). Quantification of sleep during recovery day 1 and recovery day 2 is shown in S10B and S10C Fig. Sleep stabilized in both *R23E10>Dop1R1^{RNAi}* and *R55B01>Dop1R1^{RNAi}* flies on the third day of recovery and remained elevated for several days thereafter (Fig 6A, 6B and 6E). Thus, these data indicate that the *Dop1R1* is recruited to *R23E10* and *R55B01* neurons to support waking behavior during recovery from starvation.

To gain additional insight into the physiological impact of starvation, we used live-brain imaging to evaluate the responses of *R23E10>UAS-Epac1-camps; UAS-Dop1R2^{RNAi}* and

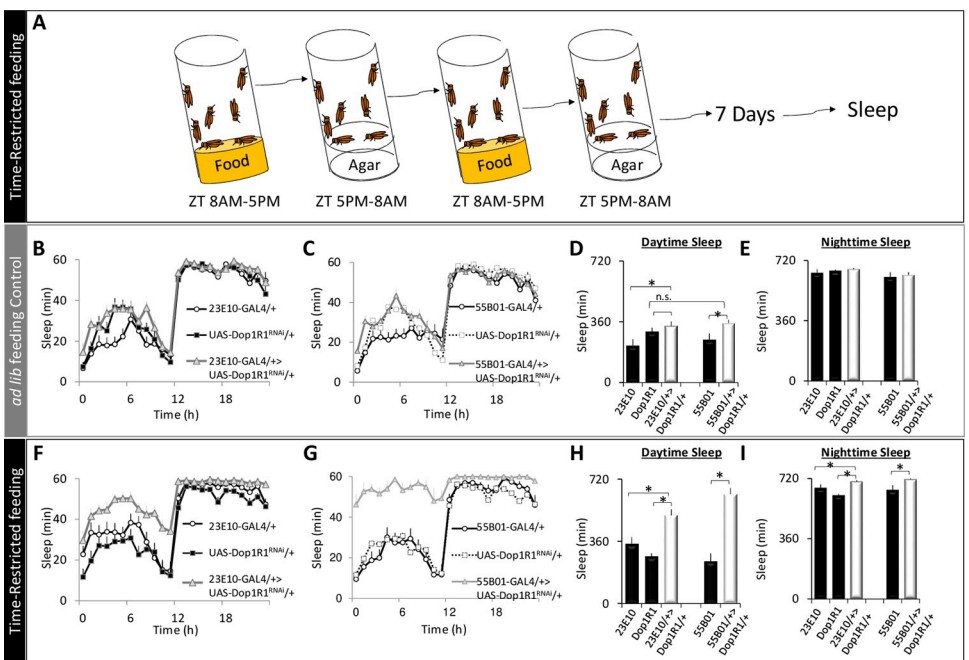

**Fig 5. Time-restricted feeding increases sleep via Dop1R1.** (**A**) Protocol for time-restricted feeding; flies were given food during ZT 8 AM to 5 PM. (**B, C**) During ad lib feeding, sleep in *R23E10>Dop1R1^RNAi* and *R55B01>Dop1R1^RNAi* flies does not differ from *R23E10/+*, *Dop1R1^RNAi/+*, or *R55B01/+* parental controls (*n* = 13–16 flies/group ANOVA for Genotype F[2,42] = 2.8, *p* = 0.07 and ANOVA for Genotype F[2,41] = 1.9, *p* = 0.15 for R23E10 and R55B01, respectively). (**D**) Daytime sleep was not altered in either *R23E10>Dop1R1^RNAi* or *R55B01>Dop1R1^RNAi* flies compared to both parental controls (ANOVA for Genotype F[2,42] = 2.8, *p* = 0.02 and F[2,41] = 2.9, *p* = 0.06, *$p < 0.05$, modified Bonferroni test). (**E**) Nighttime sleep was not changed in either *R23E10>Dop1R1^RNAi* or *R55B01>Dop1R1^RNAi* flies compared to parental controls (ANOVA for Genotype F[2,42] = 0.9, *p* = 0.40 and ANOVA for Genotype F[2,41] = 0.79, *p* = 0.46, *$p < 0.05$, modified Bonferroni test). (**F, G**) Following time-restricted feeding, sleep is increased in *R23E10>Dop1R1^RNAi* and *R55B01>Dop1R1^RNAi* flies compared to *R23E10/+*, *Dop1R1^RNAi/+*, or *R55B01/+* parental controls (*n* = 13–16 flies/group ANOVA for Genotype F[2,40] = 68.8, *p* = 9.9^eE-16 and ANOVA for Genotype F[2,35] = 38.1.8, *p* = 8.1^eE-10 for *R23E10* and *R55B01*, respectively). (**H**) Daytime sleep was significantly increased in *R23E10>Dop1R1^RNAi* and *R55B01>Dop1R1^RNAi* flies following time-restricted feeding (ANOVA for Genotype F[2,40] = 13.6, *p* = 2.96^E-05 and F[2,35] = 30.5, *p* = 2.07^E-08, *$p < 0.05$, modified Bonferroni test). (**I**) Nighttime sleep was not changed in either *R23E10>Dop1R1^RNAi* or *R55B01>Dop1R1^RNAi* flies compared to parental controls (ANOVA for Genotype F[2,40] = 11.41, *p* = 0. 0.0001 and ANOVA for Genotype F[2,35] = 8.9, *p* = 0.0007, *$p < 0.05$, modified Bonferroni test). Error bars represent SEM. Underlying data is in S5 Data and S1 Datasheet. ZT, zeitgeber time.

*R55B01>UAS-Epac1-camps; UAS-Dop1R2^RNAi* to DA under baseline and on recovery day 2 from starvation. While both *Dop1R1* and *Dop1R2* couple to *Gαs*, *Dop1R2* appears to also activate *Gαq* [65]. We hypothesized that knocking down *Dop1R2* would reduce the response of *R23E10* neurons to DA under baseline conditions and that the neurons would respond to DA following 2 days of recovery from starvation. *R23E10>UAS-Dop1R2^RNAi* flies displayed an increase in sleep under baseline conditions consistent with previous reports [19]. As seen in Fig 7A and 7B, *R23E10>UAS-Epac1-camps; UAS-Dop1R2^RNAi* neurons did not show a strong response to DA during baseline. However, during recovery from starvation *R23E10>UAS-Epac1-camps; UAS-Dop1R2^RNAi* neurons exhibited a significant increase in their response to DA suggesting a new receptor is added (Fig 7A and 7B). Next, we evaluated the effects of starvation on *R55B01* neurons. As seen in Fig 7C and 7D, neither *R55B01>UAS-Epac1-camps* nor *R55B01>UAS-Epac1-camps;UAS-Dop1R2^RNAi* responded to DA under baseline conditions. Although much smaller than that observed for *R23E10* neurons, *R55B01>UAS-Epac1-camps; UAS-Dop1R2^RNAi* neurons did respond modestly to DA after 2 days of recovery from

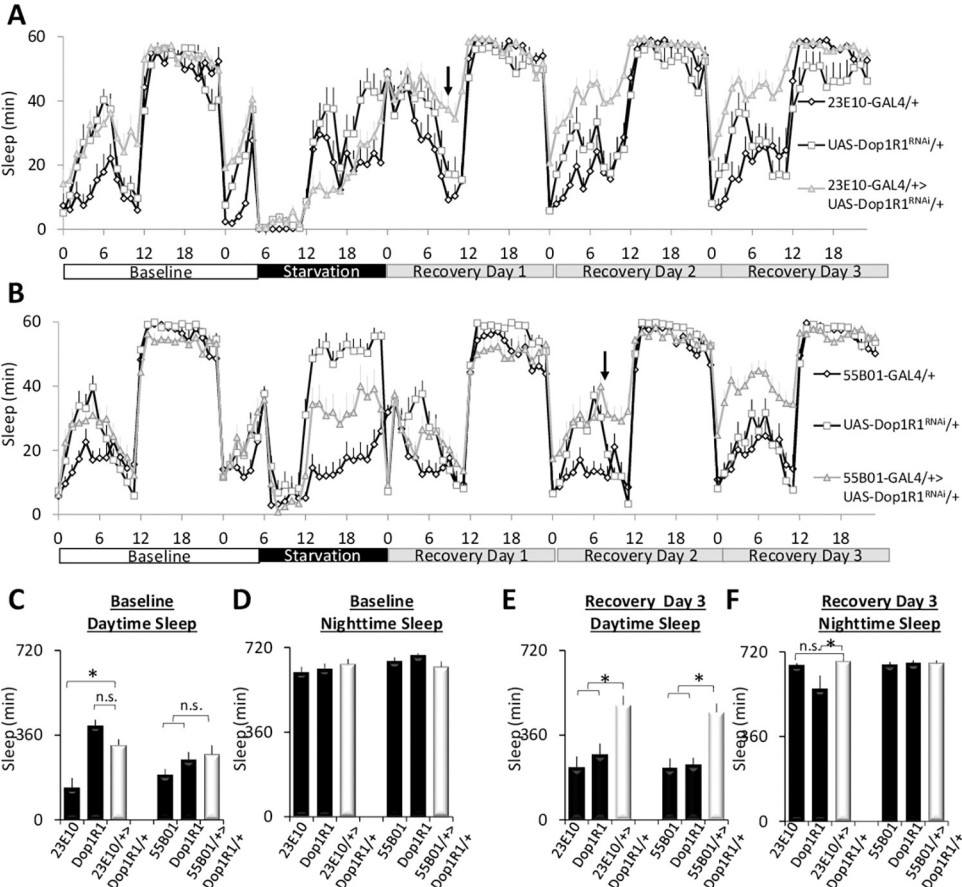

**Fig 6. Starvation alters recovery sleep and the response properties of sleep-promoting neurons.** (**A, B**) Sleep profiles in *R23E10>Dop1R1^RNAi*, *R55B01>Dop1R1^RNAi* flies, and their parental controls *R23E10/+*, *Dop1R1^RNAi/+*, and *R55B01/+* during baseline, 18 h of starvation and 3 days of recovery (*n* = 13–16 flies/group). (**C, D**) During baseline, no changes in daytime or nighttime sleep were observed in *R23E10>Dop1R1^RNAi* or *R55B01>Dop1R1^RNAi* flies compared to both parental controls (Daytime: ANOVA F$[2,36]$ = 8.1, *p* = 0.001 and F$[2,38]$ = 2.26, *p* = 0.11), (Nighttime: ANOVA F$[2,36]$ = 1.5, *p* = 0.22 and ANOVA F$[2,38]$ = 1.6, *p* = 0.20, *p* < 0.05, modified Bonferroni test). (**E**) On recovery day 3, daytime sleep is increased in *R23E10>Dop1R1^RNAi* and *R55B01>Dop1R1^RNAi* flies compared to *R23E10/+*, *Dop1R1^RNAi/+*, or *R55B01/+* parental controls (ANOVA F$[2,36]$ = 9.1, *p* = 0.0006 and F$[2,38]$ = 12.1, *p* = 8.43^E-05). (**F**) Nighttime sleep is not changed in *R23E10>Dop1R1^RNAi* and *R55B01>Dop1R1^RNAi* flies compared to *R23E10/+*, *Dop1R1^RNAi/+*, or *R55B01/+* parental controls (ANOVA F$[2,36]$ = 3.9, *p* = 0.02 and F$[2,38]$ = 0.09, *p* = 0.9 *p* < 0.05, modified Bonferroni test). Error bars represent SEM. Underlying data is in S6 Data and S1 Datasheet.

starvation (Fig 7C and 7D). While we cannot exclude a role of other DA receptors (e.g., Dopamine/Ecdysteroid receptor) for the observed changes, when viewed with the sleep experiments shown above (Figs 5 and 6), these data suggest that during recovery from starvation the constellation of DA receptors in *R23E10* and *R55B01* changes and most likely includes the recruitment of *Dop1R1*. These data provide new insights into the mechanisms used by sleep circuits to link internal states and prior waking history with sleep need.

## Discussion

In this work, we ask whether the activity of sleep-promoting dFB neurons reflects the summation of their upstream inputs in a winner-take all strategy or if they can change their own response properties to influence how environmental demands alter behavioral state. Although one possibility does not preclude the other, our data indicate that both time-restricted feeding

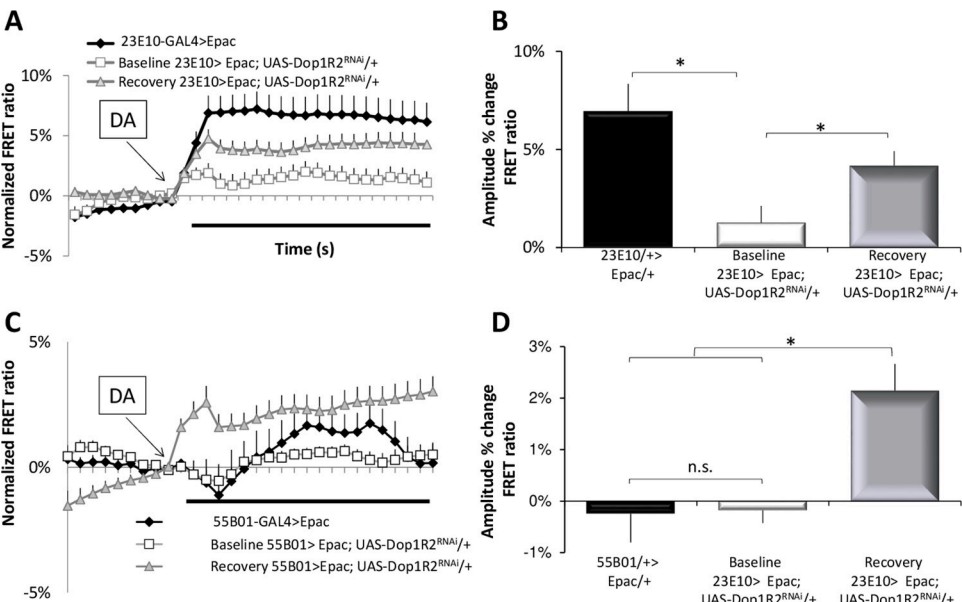

**Fig 7. Starvation alters the response properties of sleep-promoting neurons.** (A) R23E10 cells expressing the cAMP sensor Epac1-camps respond to $3 \times 10^{-5}$ M DA (black line; $n$ = 22 neurons); the response is dramatically reduced in *R23E10> UAS-Epac1-camps; Dop1R2<sup>RNAi</sup>* (white line; $n$ = 25 neurons) and partially restored during recovery from 18 h of starvation (gray line; n = 29 neurons). (**B**) Quantification of a; F[2,73] = 7.7, $p$ = 0.0009, *$p$ < 0.05, modified Bonferroni test. (**C**) The response of *R55B01> UAS-Epac1-camps* (black; $n$ = 13) and *R55B01> UAS-Epac1-camps; Dop1R2<sup>RNAi</sup>* (white; $n$ = 19) neurons to DA is very low; following recovery from starvation, the response of R55B01 neurons to DA increased (gray line; $n$ = 20). (**D**) Quantification of c; F[2,49] = 9.9, $p$ = 0.0002, *$p$ < 0.05, modified Bonferroni test. Error bars represent SEM. Underlying data is in S7 Data and S1 Datasheet. cAMP, cyclic adenosine monophosphate; DA, dopamine.

and approximately 18 h of starvation, alter the constellation of DA receptors expressed by *R23E10* and *R55B01* neurons. These results emphasize that the ability of upstream circuits to alter behavior will not only depend upon the strength of the incoming signals but also the recent historical context of *R23E10* and *R55B01* neurons themselves. In that regard, it important to note that while sleep competes with all motivated waking behavior, the proper regulation of motivated behaviors degrade in the absence of sleep [50,66–74]. Together, these observations suggest that it will not be possible to fully understand sleep's function without knowing how sleep-promoting neurons alter their functional properties to prioritize conflicting motivational states.

## The dFB is comprised of independent sets of sleep-promoting neurons

Previous studies indicate that the dFB can independently gate different aspects of sleep [26]. Moreover, the expression of the gap junction *innexin6* in *R23E10* neurons is required to gate sensory thresholds and sleep time [48]. Together, these data suggest the presence of independent sets of dFB neurons. Although we screened through a refined set of hand-selected GAL4 lines that project into the dFB in a manner similar to *104y-GAL4* and *C5-GAL4* [10,49], we only identified 1 additional GAL4 line, *R55B01*, that reliably increased sleep when crossed with *UAS-dTrpA1*. Similar results were obtained when activating the neurons using the bacterial sodium channel *UAS-NaChBac*. This latter result is relevant given that sleep is modulated by temperature and that GABAergic neurons projecting onto dFB regulate temperature-dependent changes in sleep [75,76]. We also identified dFB projecting GAL4 lines that increased waking. However, the precise role that these GAL4 lines play in waking awaits

additional inquiry. Nonetheless, these data suggest that sleep-promoting dFB neurons are limited in number. It should be noted that the extent of the overlap between *R23E10* and *R55B01* remains unclear as the *R23E10-LexA* driver does not fully recapitulate the *R23E10-GAL4* expression patterns (S5 Fig). Although it may be possible to subdivide *R23E10* and *R55B01* neurons further, our data suggest that there are at least 2 independent sets of sleep-promoting neurons that have projections into the dFB (see below for additional discussion). In mice, sleep-promoting neurons are also diverse as assessed by their role in regulating behavior and their projection patterns [77].

## Allatostatin inhibits sleep-promoting neurons

Although AstA-expressing neurons are believed to promote sleep by releasing glutamate onto *R23E10* neurons [25], our data identify a wake-promoting effect of AstA on *R23E10* neurons. Specifically, we find that knocking down either the *AstA-R1* or *AstA-R2* in *R23E10* or *R55B01* neurons using several independent RNAi lines results in a substantial increase in sleep. The observation that knocking down either the *AstA-R1* or *AstA-R2* in *R23E10* neurons increases sleep, suggests that *R23E10* neurons are under constant inhibitory tone from AstA and that both receptors play an important role in modulating sleep in untreated flies. In addition, we report that the application of 1 mM ATP to *R23E10>UAS-Epac1-camps; AstA-GAL4>P2X2* neurons reduces cAMP levels similarly to that observed with application of AstA. Although our data appear to be in conflict with that of Ni and colleagues, who provide evidence that *AstA-GAL4* provide an excitatory drive onto *R23E10* neurons, we hypothesize that the co-release of inhibitory AstA with excitatory glutamate may attenuate the overexcitement of *R23E10* neurons during high sleep drive and allow animals to maintain wakefulness in dangerous or life-threatening conditions.

In support of this hypothesis, a growing body of evidence indicates that sleep and wake regulating neurons co-release different transmitters and neuropeptides to better match sleep need with environmental demands [78]. Indeed, it has been suggested that antagonistic neurotransmitter co-release may both prevent excessive excitation of postsynaptic targets and increase the flexibility of neuronal networks [45,78,79]. For example, the co-release of GABA by histaminergic *tuberomammillary nucleus* is believed to prevent histamine-induced overexcitement of downstream circuits and thereby better support normal amounts of waking [80]. Similarly, the co-release of galanin by noradrenergic neurons is believed to prevent overexcitement of *locus coeruleus* [81]. Although the co-expression of AstA and glutamate has been established [25,82], our data reveal a new mechanism for how incompatible motivational drives can be regulated.

An alternate possibility to explain the wake-promoting effects of AstA on *R23E10* and *R55B01* neurons may be that *R23E10* and *R55B01* are, in fact, a heterogeneous sets of neurons that also include wake-promoting neurons and/or neurons whose primary role is to promote active sleep at the expense of deep sleep [83]. Active sleep is best observed following 24 h of sleep deprivation and is associated with dramatic reductions in the response to external sensory stimuli. If AstA were to inhibit a subset of *R23E10* neurons that promote active sleep, one would predict an increase in deep sleep. The diversity in the responses to DA seen in both *R23E10* and *R55B01* neurons (Fig 3F) further supports the hypothesis that these GAL4 lines express in heterogeneous set of neurons with different functions; a previous study has found similar heterogeneity in *R23E10* neurons [38]. Alternatively, it is possible that AstA does not inhibit the *R23E10* neurons per se but rather alters temporal spike patterns to favor active sleep or deep sleep in a manner similar to that observed in clock neurons [84]. Indeed, AstA has been shown to alter the spike-timing precision of mechanoreceptor afferents in *Carcinus*

*maenas* and *Galanin*, alters spontaneous spike firing in rodents [85,86]. These possibilities will be explored in future studies.

It is important to note that neuropeptides are notoriously pleiotropic and also work as neurohormones [44]. Thus, it is possible that additional sleep- and wake-promoting circuits downstream of *AstA-GAL4* or *65D05-GAL4* will be found elsewhere in the brain. It is interesting to note that neurons expressing the mouse galanin, which is functionally similar to AstA, also regulate conflicting behaviors [77]. Specifically, galanin-expressing neurons that project to the *tuberomammillary nucleus* promote sleep while galanin-expressing neurons that project to the medial amygdala promote parental behaviors [77,87,88]. Thus, it will be important for future studies to discern how AstA and galanin circuits regulate competing activities in other circuits.

### *R23E10* neurons differentially encode arousal signals

Increased sleep drive may be maladaptive in many circumstances since falling asleep could place the individual in danger of immediate physical harm [7,8]. Increased sleep also competes with important waking behaviors such as foraging, eating, and mating [3,4]. However, sleep plays a critical role in learning and memory, supports adaptive behavior, and facilitates creative insight [9–12]. How do sleep-promoting neurons distinguish between competing drives? Our live-brain imaging data suggest the intriguing possibility that *R23E10* neurons can distinguish between different types of waking and change their response properties accordingly. Specifically, *R23E10* neurons selectively <u>decrease</u> their response to the wake-promoting effects of DA following conditions that promote plasticity. In contrast, *R23E10* neurons <u>increase</u> their response to the wake-promoting effects of AstA during conditions when sleep drive is high but the expression of sleep might be dangerous (e.g., during sleep deprivation). Finally, *R23E10* neurons appear to contain a memory trace of starvation as evidenced by the increase inhibitory tone conveyed by the recruitment of *Dop1R1*. It should be emphasized that we did not use TTX when evaluating the response properties of *R23E10* neurons to different types of waking. As a consequence, it is possible that the difference in the intrinsic release of AstA and DA during these manipulations modified the response of dFB neurons. Such a possibility will be evaluated in future studies. Nonetheless, *R23E10* neurons are plastic and can utilize AstA and DA in very different ways to favor specific behavioral outcomes.

As mentioned, increasing inhibitory signals onto sleep-promoting neurons during sleep deprivation may be an adaptive response that allows animals to stay awake despite increasing sleep drive. Indeed, lesioning galanin neurons in the preoptic area were found to reduce sleep rebound in zebrafish [89] and mice [90]. However, AstA is also reported to be a satiety signal that limits feeding [27,32,37,91]. The effects of AstA on feeding seem to be in conflict with the studies linking sleep deprivation to increased food intake [92–95]. Nonetheless, recent reports have identified complex interactions between hunger and satiety signals that may be nutrient specific and can be modified by Hebbian plasticity [32,34,40,44]. Thus, it is reasonable to expect that sleep deprivation must differentially activate neurons regulating a variety of motivated behaviors including satiety and sleep drive. Indeed, these data indicate that sleep deprivation may be harnessed as a tool to further explore how internal states impact neuronal circuits regulating conflicting goal-directed behaviors [30].

### *R23E10* neurons support long-term memory

Interestingly, the ventral fan-shaped body (vFB) promotes sleep and regulates the activity of Dopaminergic aSP13 neurons (DAN-aSP13s) to consolidate long-term memory as assessed using courtship conditioning [59]. Surprisingly, activating *R23E10* neurons does not alter the

activity of DAN-aSP13s neurons suggesting that the dFB may not be involved in courtship memory [59]. It should be noted, however, that *R23E10* neurons regulate the activity of other sets of arousal promoting neurons including the MV1 Dopaminergic neurons and octopaminergic arousal neurons [25,96]. These data suggest that the dFB and vFB may play distinct roles in different memory assays. Nonetheless, our data clearly indicate that *R23E10* neurons change their response properties following training that induces LTM but not massed training. Importantly, the changes in *R23E10* neurons only return to baseline 48 h after training. It is possible that *R23E10* neurons play an important, yet indirect, role in courtship memory. That is, activating *R23E10* neurons strongly suppresses the response to external sensory stimuli [48,83]. Thus, the activity of *R23E10* neurons may protect sleep by limiting the opportunity for external stimuli to wake the animal up. Under this scenario, dFB and vFB neurons would work in concert to carry out sleep functions.

In addition, we find that courtship conditioning increases the activity of *R23E10* neurons as assessed by *CaLexA*. Consistent with this observation, spaced-training reduced the inhibitory effects of DA on *R23E10* neurons. Importantly, massed-training, which only induced short-term memory, does not alter the response of *R23E10* neurons to DA. A previous report has shown that the ectopic expression of *Dop1R1* in dFB neurons reduces sleep and fails to rescue courtship conditioning memory in *Dopamine transporter* mutants (*fmn*) [36]. Together, these data emphasize that reducing Dopaminergic signaling to the dFB neurons is important for long-term memory following courtship conditioning.

## Internal state remodels dopamine receptors in sleep-promoting neurons

The arousal promoting properties of DA projections to the dFB are well established [19,25,35,36,61]. However, which DA receptor is responsible for the increased waking remains controversial. Initial results, using the ectopic expression *Dop1R1* and live-brain imaging implicated *Dop1R1* [35,36]. Nonetheless, RNAi knockdown of *Dop1R1* in dFB neurons did not alter sleep [35]. A subsequent study demonstrated that activation of *Dop1R2* results in a transient hyperpolarization of dFB neurons followed by a lasting suppression of excitability lasting minutes; knocking down *Dop1R2* but not *Dop1R1* altered sleep [19]. Our data replicate previous RNAi studies that have failed to observe a change in sleep following the expression of *Dop1R1*^RNAi^ in dFB neurons in untreated flies. Indeed, no changes in FRET signal were observed upon the bath application of DA to untreated *R23E10>UAS-Epac1-camps; Dop1R2*^RNAi^ or *R55B01>UAS-Epac1-camps; Dop1R2*^RNAi^ flies further supporting the role of *Dop1R2* in baseline sleep regulation.

However, in the context of time-restricted feeding and starvation, knocking down *Dop1R1* in *R23E10* or *R55B01* neurons resulted in substantial increases in sleep. We hypothesize that both time-restricted feeding and starvation place energy-saving sleep drive in conflict with a metabolic signal that facilitates motivated waking behavior. Increased wake drive would allow the animals to be more alert and productive during their primary wake period. Under these circumstances, *Dop1R1* would be recruited to provide additional inhibitory tone to sleep-promoting *R23E10* and *R55B01* neurons during the light period. Interestingly, starvation results in the recruitment of the *pigment-dispersing factor receptor* (*Pdfr*) into wake-promoting *large ventrolateral neurons* (*lLNvs*) to facilitate waking at the end of the biological day [3]. Whether other sleep- and wake-promoting circuits are regulated in a similar fashion is an open question that will require additional inquiry.

## Conclusions

Given the important roles that DA and AstA play in regulating motivated behaviors, including sleep, we studied their impact on a subset of sleep-promoting neurons. Our data reveal that

AstA is wake-promoting and likely serves to maintain waking during periods of high sleep drive. In addition, our data reveal that time-restricted feeding or 18 h of starvation recruits the *Dop1R1* to sleep-promoting neurons to maintain wakefulness during the day. These results are consistent with our previous data showing that the wake-promoting lLNvs recruit the *Pdfr* following sleep loss to facilitate waking and that wing-cut reactivates a developmental sleep circuit [3,97]. The ability of sleep- and wake-promoting neurons to alter their own physiology, including the recruitment of a new receptor, provides important clues into sleep regulation and function.

## Supporting information

**S1 Fig. Changes in sleep in parental controls following 6 h at 31˚C.** (**A**) The % change from baseline sleep at 25˚C seen following switch to 31˚C between 9 AM and 3 PM. (B) Confocal image of *R23E10>UAS-Epac1-camps*. Underlying data is in S1 Datasheet.
(TIF)

**S2 Fig. *R23E10* neuropeptide RNAi screen for Daytime Sleep.** (**A**) Individual cAMP responses of *R23E10* neurons expressing *UAS-Epac1-camps* shown as % change in FRET ratio during exposure to crustacean cardioactive peptide (CCAP), *Drosophila* myosuppressin (DMS), allatostatin C (astA C), proctolin, TPAEDFMRFamide, corticotropin-releasing factor-like diuretic hormone 44 (DH44), Tachykinin 1, Tachykinin 3, short neuropeptide F (sNPF), adipokinetic hormone (AKH), corazonin, and melatonin. (**B**) Daytime sleep in female 5-day-old flies expressing RNAi lines for the depicted neuropeptide receptors using *R23E10-GAL4* and their parental controls (*n* = 14–16 flies/genotype). To be significant, the experimental lines must be significantly different from both the GAL4/+ (red line) and the UAS/+ (white bar) parental controls: FR: *FMRFamide Receptor*, DHRRR2: *Diuretic hormone 44 receptor 2*, DMSR1: *Myosuppressin receptor 1*, DHRRR1: *Diuretic hormone 44 receptor 1*, cchamideR: *CCHamide-1 receptor*, NepyR: *RYamide receptor*, Capar: *Capability receptor*, Pk2R1: *Pyrokinin 2 receptor 1*; CCKLR: *Cholecystokinin-like receptor at 17D1A*, DH31: *Diuretic hormone 31*, AstC-R1: *Allatostatin C receptor 1*, TkR99D: *Tachykinin-like receptor at 99D*, MsR2: *Myosuppressin receptor 2*, Lkr: *Leucokinin receptor*, CrzR: *Corazonin receptor*, Proc: *Proctolin receptor*. Red line is to facilitate comparisons with *R23E10/+* parental control. (**C**) RNAi knockdown of the AstA-R1 receptor in *R23E10* neurons did not affect the cAMP response to DA. Error bars represent SEM. Underlying data is in S1 Datasheet.
(TIF)

**S3 Fig. Expressing *AstA-R1-RNAi* or *AstA-R2-RNAi* in *R23E10* neurons alters sleep architecture.** (**A, B**) Nighttime sleep bout duration is increased in *UAS-Dcr2, R23E10-GAL4/+> AstA-R1RNAi/+*, and *UAS-Dcr2,R23E10-GAL4/+> AstA-R2RNAi/+* experimental flies compared with *UAS-Dcr2, R23E10-GAL4/+, AstA-R1RNAi/+*, and *AstA-R2RNAi/+* parental controls (*n* = 16/condition, \**p* < 0.05, modified Bonferroni test). (**C, D**) Sleep latency is shortened in *UAS-Dcr2, R23E10-GAL4/+> AstA-R1RNAi/+*, and *UAS-Dcr2,R23E10-GAL4/+> AstA-R2RNAi/+* experimental flies compared with *UAS-Dcr2, R23E10-GAL4/+, AstA-R1RNAi/+*, and *AstA-R2RNAi/+* parental controls (*n* = 16/condition, \**p* < 0.05, modified Bonferroni test). (**E**) Total sleep is increased when levels of AstA-R1 is decreased in *R23E10* sleep-promoting neurons using 3 additional independent RNAi lines (*n* = 16/condition, \**p* < 0.05, modified Bonferroni test). (**F**) Sleep is increased in *AstA-GAL4>UASdTrpA1* and *65D05-LexA>LexAopdTrpA1* flies at 31˚C compared with siblings maintained at 25˚C; parental controls did not show an increase in sleep at 31˚C (*n* = 14–16/condition and genotype, \**p* < 0.05,

modified Bonferroni test). Error bars represent SEM. Underlying data is in S1 Datasheet.
(TIF)

**S4 Fig. Screen of Janelia-GAL4 lines that express in the dorsal fan-shaped body in a pattern similar to that observed with *104y-GAL4* and *C5-GAL4*. (A)** Despite similar anatomical profiles, most dorsal fan-shaped body drivers do not reliably impact sleep when expressing *UAS-Transient receptor potential cation channel A1* and raising the temperature to 31˚C ($n = 14–16$ flies/genotype). The data for *R55B01>dTrpA* are the same as in Fig 3A. **(B)** *R23E10>UAS-NaChBac* and *R55B01>UAS-NaChBac* sleep more than *R23E10/+*, *UAS-NaChBac/+*, and *R55B01/+* parental controls. Error bars represent SEM. Underlying data is in S1 Datasheet.
(TIF)

**S5 Fig. Anatomy of *R23E10* and *R55B01* neurons. (A)** Representative confocal stack focusing on the area containing cell bodies of a *R23E10-LexA>LexAop-GFP, R55B01-GAL4>UAS-RFP* fly brain stained with anti-GFP antibody, **(A')** anti-RFP antibody (magenta) and **(A")** a merged image. Yellow arrows on the merge image indicate cells that express both GFP and RFP. **(B)** Quantification of the number of cells expressing only GFP, only RFP or both GFP and RFP. **(C)** Representative confocal stack focusing on the area containing the cell bodies of a *R23E10-LexA>LexAop-GFP, R23E10-GAL4>UAS-RFP* fly brain stained with anti-GFP antibody, **(C')** anti-RFP antibody (magenta), and **(C")** a merged image. **(D)** Quantification of the number of cells expressing only GFP, only RFP or both GFP and RFP. **(E)** Representative confocal stack of a using *R55B01-GAL4>UAS-GFP* brain stained with anti-GFP antibody, **(E')** anti-AstA antibody (magenta), and **(E")** a merged image. Error bars represent SEM. Underlying data is in S1 Datasheet.
(TIF)

**S6 Fig. Expressing *AstAR1^RNAi^* or *AstAR2^RNAi^* R55B01 neurons alters sleep architecture.** Sleep parameters in *UAS-Dcr2, R55B01-GAL4/+> AstA-R1^RNAi^/+, UAS-Dcr2, R55B01-GAL4/+> AstA-R2^RNAi^/+* experimental flies and both *UAS-Dcr2, R55B01-GAL4/+* and *AstA-R1^RNAi^/+, AstA-R2^RNAi^/+* control flies for: **(A, B)** total sleep, **(C, D)** nighttime sleep bout duration, **(E, F)** nighttime sleep latency, **(G, H)** daytime sleep bout duration, and **(I, J)** counts/waking minute; ($n = 16$/condition, $^*p < 0.05$, modified Bonferroni test). Error bars represent SEM. Underlying data is in S1 Datasheet.
(TIF)

**S7 Fig. Response of *R23E10* neurons to Dopamine is not changed in older flies.** Response of *R23E10* neurons in young (0–1 day old), adult (6–8 day old), and old (30–38 day old) flies in response to Dopamine ($3e^{-5}$ M) ($n = 15, 14, 6$ cells). The data are expressed as amplitude % change; data for young and adult flies are taken from data in Fig 4B. Error bars represent SEM. Underlying data is in S1 Datasheet.
(TIF)

**S8 Fig. Space-trained flies vs. naïve controls.** Space-trained flies vs. naïve controls. The reduction of DA responses remained significant 24 h after the end of the training but not 48 h post-training. A Massed training courtship protocol that does not induce LTM had no significant effect on amplitude of DA response in *R23E10* neurons; 4 h of sleep deprivation did not alter DA responses in R23E10 neurons ($n = 12–25$ cells, $^*p < 0.05$). Error bars represent SEM. Underlying data is in S1 Datasheet.
(TIF)

**S9 Fig. Time-restricted feeding changes sleep-promoting neurons. (A)** Sleep profiles in *R23E10>Dop1R1^RNAi^*, flies and their parental controls *R23E10/+*, and *Dop1R1^RNAi^/+* during

baseline, 7 days of time-restricted feeding and 5 days of recovery ($n$ = 24–26 flies/group); gray arrow indicates disruption in data collection. (**B**) Daytime sleep is increased following time-restricted feeding in *R23E10>Dop1R1^RNAi* compared to both parental controls. A Genotype (2) x Time (13) ANOVA revealed a Genotype X Time interaction: $F_{[2,24]}$ = 5.5, $p$ = 9.99^E-16; \*$p$ < 0.05, modified Bonferroni test. (**C**) Sleep profiles in *R55B01>Dop1R1^RNAi* flies and their parental controls *R55B01/+*, and *Dop1R1^RNAi/+* during baseline, 7 days of time-restricted feeding and 5 days of recovery ($n$ = 24–30 flies/group) (Dop1R1-RNAi flies are the same as in A, B); gray arrow indicates disruption in data collection. (**D**) Daytime sleep is increased following time-restricted feeding in *R23E10>Dop1R1^RNAi* compared to both parental controls. A Genotype (2) x Time (13) ANOVA revealed a Genotype X Time interaction: ANOVA $F_{[2,24]}$ = 7.87, $p$ = 9.99^E-16; \*$p$ < 0.05, modified Bonferroni test. Underlying data is in S1 Datasheet.
(TIF)

**S10 Fig. Starvation alters recovery sleep.** (**A**) During starvation, sleep in *R23E10>Dop1R1^RNAi*, *R55B01>Dop1R1^RNAi* flies is not consistently above or below *R23E10/+*, *Dop1R1^RNAi/+* or *R55B01/+* parental controls ($n$ = 13–16 flies/group; ANOVA $F[2,36]$ = 2.2, $p$ = 0.12 and ANOVA $F[2,38]$ = 15.4, $p$ = 1.2^E-05, \*$p$ < 0.05, modified Bonferroni test). (**B**) Daytime sleep is not increased in *R23E10>Dop1R1^RNAi* or *R55B01>Dop1R1^RNAi* flies compared to both parental controls *R23E10/+*, *Dop1R1^RNAi/+*, or *R55B01/+* on recovery day 1 (ANOVA $F[2,36]$ = 4.8, $p$ = 0.02 and ANOVA $F[2,38]$ = 0.5, $p$ = 0.58, \*$p$ < 0.05, modified Bonferroni test). (**C**) Sleep is increased in *R23E10>Dop1R1^RNAi* and *R55B01>Dop1R1^RNAi* flies compared to *R23E10/+*, *Dop1R1^RNAi/+*, or *R55B01/+* parental controls on recovery day 2 (ANOVA for Genotype $F[2,36]$ = 14.25, $p$ = 2.75^E-05 and ANOVA for Genotype $F[2,38]$ = 8.35, $p$ = 0.0009 for *R23E10* and *R55B01*, respectively. Underlying data is in S1 Datasheet.
(TIF)

**S11 Fig. Expressing *Dop1R2-RNAi* in *R23E10* neurons increases sleep.** (**A**) Sleep is increased in *R23E10>Dop1R2^RNAi*, flies compared to parental controls ($n$ = 16 flies/group; 3 (genotype) X 24 (hour) repeated measures ANOVA reveals a significant interaction $F_{[46,1012]}$ = 7.15, $p$ = 0.1.4^E-13). Underlying data is in S1 Datasheet.
(TIF)

**S1 Data. Data underlying Fig 1.**
(XLSX)

**S2 Data. Data underlying Fig 2.**
(XLSX)

**S3 Data. Data underlying Fig 3.**
(XLSX)

**S4 Data. Data underlying Fig 4.**
(XLSX)

**S5 Data. Data underlying Fig 5.**
(XLSX)

**S6 Data. Data underlying Fig 6.**
(XLSX)

**S7 Data. Data underlying Fig 7.**
(XLSX)

**S1 Datasheet. Data underlying S1–S11 Figs.**
(XLSM)

**S1 Text. Supplementary Material and Method.**
(DOCX)

## Acknowledgments

We thank Gerald Rubin, Arnim Jenett, Jin Wang, Ori Shafer, and Paul Taghert for sharing reagents and flies. The confocal facility is supported by NIH shared instrument grant S1OD21629-01A1.

## Author Contributions

**Conceptualization:** Stephane Dissel, Markus K. Klose, Bruno van Swinderen, Paul J. Shaw.

**Data curation:** Stephane Dissel, Markus K. Klose, Lijuan Cao, Melanie Ford, Erica M. Periandri, Joseph D. Jones, Zhaoyi Li, Paul J. Shaw.

**Formal analysis:** Stephane Dissel, Markus K. Klose, Melanie Ford, Erica M. Periandri, Joseph D. Jones, Zhaoyi Li, Paul J. Shaw.

**Funding acquisition:** Paul J. Shaw.

**Investigation:** Stephane Dissel, Melanie Ford, Erica M. Periandri, Joseph D. Jones, Zhaoyi Li, Paul J. Shaw.

**Methodology:** Stephane Dissel, Markus K. Klose, Bruno van Swinderen, Melanie Ford, Zhaoyi Li, Paul J. Shaw.

**Resources:** Bruno van Swinderen, Paul J. Shaw.

**Supervision:** Stephane Dissel, Paul J. Shaw.

**Validation:** Stephane Dissel, Erica M. Periandri, Paul J. Shaw.

**Visualization:** Stephane Dissel, Paul J. Shaw.

**Writing – original draft:** Stephane Dissel, Markus K. Klose, Bruno van Swinderen, Lijuan Cao, Paul J. Shaw.

**Writing – review & editing:** Stephane Dissel, Markus K. Klose, Bruno van Swinderen, Lijuan Cao, Paul J. Shaw.

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
