## [Editor Report · Decision Letter 0]

3 Dec 2021

Dear Dr Shaw, 

Thank you for submitting your manuscript entitled "Sleep promoting neurons remodel their response properties to calibrate sleep drive with environmental demands" for consideration as a Research Article by PLOS Biology. I apologize for our delay in sending you an initial decision - we have had a bit of a hectic week catching up after the Thanksgiving Holiday. 

Your manuscript has now been evaluated by the PLOS Biology editorial staff as well as by an academic editor with relevant expertise and I am writing to let you know that we would like to send your submission out for external peer review.

Once your full submission is complete, your paper will undergo a series of checks in preparation for peer review. Once your manuscript has passed the checks it will be sent out for review. To provide the metadata for your submission, please Login to Editorial Manager (https://www.editorialmanager.com/pbiology) within two working days, i.e. by Dec 07 2021 11:59PM.

If your manuscript has been previously reviewed at another journal, PLOS Biology is willing to work with those reviews in order to avoid re-starting the process. Submission of the previous reviews is entirely optional and our ability to use them effectively will depend on the willingness of the previous journal to confirm the content of the reports and share the reviewer identities. Please note that we reserve the right to invite additional reviewers if we consider that additional/independent reviewers are needed, although we aim to avoid this as far as possible. In our experience, working with previous reviews does save time. 

If you would like to send previous reviewer reports to us, please email me at lsmith@plos.org to let me know, including the name of the previous journal and the manuscript ID the study was given, as well as attaching a point-by-point response to reviewers that details how you have or plan to address the reviewers' concerns. 

Given the disruptions resulting from the ongoing COVID-19 pandemic, please expect some delays in the editorial process. We apologize in advance for any inconvenience caused and will do our best to minimize impact as far as possible.

Kind regards,

Lucas

Lucas Smith

Associate Editor

PLOS Biology

lsmith@plos.org

---

## [Decision Letter · Decision Letter 1]

18 Jan 2022

Dear Paul,

I am writing on behalf of my colleague Lucas Smith, who is currently on paternity leave. Thank you for submitting your manuscript "Sleep promoting neurons remodel their response properties to calibrate sleep drive with environmental demands." for consideration as a Research Article at PLOS Biology. Your manuscript has been evaluated by the PLOS Biology editors, by an Academic Editor with relevant expertise, and by three independent reviewers. Please accept our apologies for the delay in communicating the decision below to you.

In light of the reviews (below), we will not be able to accept the current version of the manuscript, but we would welcome re-submission of a much-revised version that takes into account the reviewers' comments. We cannot make any decision about publication until we have seen the revised manuscript and your response to the reviewers' comments. Your revised manuscript is also likely to be sent for further evaluation by the reviewers.

We expect to receive your revised manuscript within 3 months. 

**IMPORTANT - SUBMITTING YOUR REVISION**

Your revisions should address the specific points made by each reviewer. They all make (numerous) different points, but nothing seems to require an enormous amount of new experimental work (and often just text changes) - except for the DopR1 RNAi experiment.

Please submit the following files along with your revised manuscript:

*Re-submission Checklist*

*Published Peer Review*

*PLOS Data Policy*

*Blot and Gel Data Policy*

Sincerely,

Gabriel Gasque on behalf of 

Lucas Smith

Associate Editor

PLOS Biology

lsmith@plos.org

REVIEWS:

Reviewer #1: In this manuscript, Dissel and colleagues dissect the response of sleep-promoting neurons in Drosophila to allatostatin and dopamine, in both normal conditions and under various circumstances that modulate sleep need. They find in particular that dopamine responses are modulated by various starvation paradigms, which also modulate sleep and propose this is due to the upregulation of a specific dopamine receptor that is not normally required for dopamine sensitivity in these neurons under baseline conditions. 

The paper is somewhat of a hodge-podge of observation about dFB sleep regulating neurons, spanning allatostatin inputs, a novel sleep-promoting dFB neuron, dopamine inputs, starvation, long-term memory, neural plasticity of receptor expression. This leads to a complicated set of loosely-connected results. However, I am favorably inclined towards this paper, with several interesting observations worthy of publication. I have a few small comments and one bigger comment.

To my mind, the most interesting finding is the sculpting of dopamine receptor subtypes in the dFB neurons and how this alters the integration of sleep responses to starvation. However, the conclusion is based on RNAi behavioral experiments against Dop1R1, showing an impact on sleep in some but not other starvation paradigms (Figure 5+6), combined with cAMP imaging experiments during starvation with RNAi against a different dopamine receptor Dop1R2 (Figure 7), showing that this has no effect on the cAMP signal to dopamine under conditions where starvation-induced sleep is dopamine dependent. The data is consistent with their interpretation, but showing that RNAi against Dop1R1 abrogates the effect would be the most direct demonstration, and I am not clear why this wasn't included.

Other specific comments: 

1) the abstract says, "the Drosophila sleep homeostat (dfB)", but this is both confusing and misleading. dFB stands for "dorsal fan-shaped body neurons", not "sleep homeostat" as implied somehow here. More critically, calling these neurons the sleep homeostat is a bit of an overstatement-- the best evidence is that these are an important output arm of the sleep homeostat, but the evidence is out whether the whole sleep homeostatic mechanism is embodied by these neurons. 

2) The P2X2 data showing a connection between AstA-neurons and R23E10 neurons could still be an indirect, or even a long-range, connection, given the slow timescale of the reporter. Does the recently published fly EM map provide any evidence of direct connections? 

3) Knocking down AstAR2 in R55B01 neurons increases waking activity compared to the control strains. This could suggest an alternative interpretation of a specific sleep effect-- i.e. hyperlocomotion, that should at least be acknowledged in the main text when disucssing Figure S6. 

4) line "we first evaluated the response properties or R23E10 neurons) should read "of"

5) In the Discussion, comments are made about the data being in conflict with Ni and colleagues, without elaboration. This point should be made clearer-- what was the Ni result specfically, how is your result different? 

6) Discussion: Galanin is mentioned as the homolog of AstA multiple times, when once is enough. It is also a little inaccurate to call it the "mammalian" homolog, when it is in all vertebrates. It also confuses the connection between AstA and Galanin a little bit-- by traditional homology, these peptides look nothing alike. The claim is based on some similarity between the Galanin-receptor and the Allatostatin receptor, but other GPCRs also are similar-- are those orthologs, too? it is fine to claim functional similarities of these inhibitory neuropeptides but let's make the terms of this observation more explicit

7) Discussion: "lesioning Galanin neurons in the preoptic area reduces sleep rebound following sleep loss in mice". The impact of galanin loss on sleep rebound was shown first by Reichert 2019 in the Zebra Fish

8) Conclusion: "Our data reveal that the co-release of AstA is wake-promoting". This is overstated-- the co-release is presumeably the hypothesized one with glutamate, but this is not clearly and directly demonstrated in any experiment in this paper. 

9) Figure 3A-- the data appears identical to the screen data presented in Figure S4. This should be acknowledged as such, as otherwise it implies the screen result was independently confirmed in a biological replicate, when it does not appear to have been so. It should be done as a separate confirmatory result from the screen in S4.

10) Figure 4 lacks stats, which are buried in the Figure S7, but these are critical for the interpretation of the curves. Couldn't a repeated measures stat capture this in the same panel? Or perhaps the graphs of the supplement can be made into smaller insets, especially for the key findings?

11) Figure 7D, the 55801/+>Epac/+ bar is small and slightly negative, but this doesn't quite match the data shown in Figure 7C, which has a short dip followed by data that are mostly positive. This disconnect makes me concerned about how the time series are being normalized and summed/averaged to generate the bar graphs that form the basis of the data analysis, here and in other figures measuring the cAMP levels. Would a time-series repeated measures stat serve better here? 

12) In general, the standard for depicting data in figures nowadays is to show all the datapoints, not just a bar graph with SEM. That allows for a fuller representation of the data, highlights the Ns and variance involved, etc. I would highly recommend and strongly prefer such plots for all the bar graph data in the manuscript. e.g. Kempf et al., 2019. or Grubbs et al 2020 for a sleep-paper model systems in PloS Biology example. 

Reviewer #2: The authors examined the role of allatostatin-A (AstA) on sleep-promoting R23B10/R55B01 positive dorsal fan-shaped body (dFB) neurons of Drosophila and found AstA decreased intracellular cyclic AMP (cAMP) level (Fig 1). Since knocking down of either AstA-R1 or AstA-R2 increased sleep (Fig 2, Fig 3GD), they proposed AstA promotes wakefulness by inhibiting dFB neurons, although the activation of AstA positive neurons by AstA-GAL4>dTRP-A1 promoted sleep (Fig S3C). They also examined the changes in the cAMP responses of R23B10/R55B01 positive dFB neurons to AstA and dopamine (DA) under different conditions, which affect sleep behaviors (Fig 4). Various changes were observed and apparently the responses to AstA and DA were independently and context-dependently controlled. 

The authors also found that knocking down of Dop1R1 in R23B10/R55B01 positive dFB neurons, which did not affect sleep amount under ordinary condition, increased sleep under restricted feeding protocol (Fig 5). Knocking down of DopR1 in dFB also changed the sleep behavior during the recovery period from starvation (Fig 6). They showed the responses of dFB neurons with Dop1R2 knockdown to DA were changed from starvation to recovery (Fig 7). 

With all these data, the authors concluded that dFB neurons changed their state, i.e. responsiveness to AstA and DA, which may be the basis for the distinction of the different type of sleep needs.

I enjoyed reading the manuscript and found it interesting and informative, although there are some inconsistency and insufficiency.

Major questions.

1. The biggest changes in sleep need and arousal threshold occur between day and night. I think it is necessary to examine the changes of dFB neurons at different time of the day. At least, the time of the experiments should be clearly stated.

2. Were the experiments of Fig 4 and Fig 7 performed with TTX? Otherwise, it is possible that the difference in the intrinsic release of AstA and DA modified the response of dFB neurons to them. This possibility should be clearly stated.

3. In addition to above, although this might be technically difficult, the basic intracellular cAMP level of dFB neurons before the application of AstA and DA may be different according to the context in Fig 4. This difference affects the magnitude of AstA and DA responses differently, since the direction of the response is reverse, i.e. decrease and increase. This should be also discussed.

4. The interpretation of Fig 6 is too speculative. Since there is no change in DA response after starvation in control flies as shown in Fig 4h, the mechanism would be different from Dop1R1 induction. The authors should demonstrate actually induction of Dop1R1 by immunohistochemistry or other methods. Fig 7 is also pretty incomplete. Since, there is no change in the DA response of the control fly after starvation, the DA response change in the flies during recovery period and that of the Dop1R2 RNAi flies after starvation should be included. In addition, since the magnitude of the effect of time restricted feeding in Fig 5 is very big, it is better to include the live imaging data of Dop1R1 knockdown before and after TRF. 

Minor points

1. Please state the sex of the flies used clearly, although the authors wrote there were no big sexual dimorphism.

2. Please include the more discussion about Fig 4. For example, I do not understand why the response to DA did not change after sleep deprivation in Fig 4d.

3. The abstract should include Ast-A and DA which is helpful to the authors.

4. The phrase "Rather, the dFB neurons themselves can determine their response to the activity from upstream circuits" in the abstract does not make sense to me. I think the authors showed that dFB neurons changes their responsiveness from the previous inputs to them from other neurons, but not they sense the sleep need for themselves.

Reviewer #3: In this manuscript, Dissel et al examine examine how the dFB and input neurons are regulated in response to a number of different stresses. The manuscript focuses on two neuromodulations, AstA and DA, that impact R23E10 neurons that innervate the dFB. A major strength of this manuscript is that it provides cellular-level resolution of how sleep neurons are regulating depending on experience. This aspect of the paper is likely to generate broad interest because conveys the need to understand mechanisms regulating sleep in a number of different conditions. A number of experimental concerns are described below, some of which can be addressed through written clarification). There is a heavy reliance on ex vivo preparations, which may not reflect the brain function in an intact animal. In addition, the manuscript is written in a descriptive manner, and streamlining the writing and figures may help to convey the conceptual advances and increase the impact of the manuscript. 

1. I find the logic laid out in the abstract suggesting the dFB determine responses to be confusing. The way it is written, it seems to confer cognitive capacity on a set of neurons . The data suggests these neurons are integrators of information of other areas, impacting the behavioral output. In the third paragraph of the introduction the logic is laid out, but it is not clear what the alternative is to being impacted by upstream regulators.

2. Why does Epac1-camps label fewer cells than GFP? Are the eight a different subset? Is the difference due to sensitivity/different levels of R23E10? It would be useful to show these neurons.

3. In 1i, Asta1-RNAi nearly abolishes FRET, but both AstARs impact sleep in Figure 2. Please explain and/or include the effects of AstA-R2-RNAi on Epac1 FRET.

4. If data were acquired where ATP is washed out in 1L this is important to show.

5. Fig 2I does not fully test the hypothesis that AstA signals through the identified G-proteins because there are likely many receptors using G-proteins in those cells. I suggest toning down thisstatement. Also, it's not clear what the alternative mode of action would be for AstaRs since they are G-protein coupled.

6. There is a large reliance on ex vivo imaging through. It is important to highlight the limitations of this or validate with an in vivo preparation. In addition, given the role of AstA in circadian regulation, emphasizing the timing of experiments throughout the manuscript would be helpful.

7. A weakness is the focus on AstA and DA, but there are likely additional contributors to the regulation of R23E10 neurons.

8. The TRF protocol used in figure 5 lists the restriction from 8AM-5PM. This should be in ZT, but it is also unclear why this timeframe was chosen, since it does not match the light dark cycle or any obvious component of the sleep cycle. I also question the idea that TRF results in substantial metabolic challenge. An alternative way of interpreting TRF is that misalignment is a metabolic challenge, and the metabolic stress is reduced during TRF. The robustness of the effects in 5F-I is very compelling. It would be informative to have data on sleep during the TRF protocol.

9. This may be semantic, but I think it's conceptually important to lay out whether Dop1R1 'supports waking behavior during recovery from starvation' or inhibits homeostatic recovery.

Minor: 

1. In the abstract, the (dFB) abbreviation is confusing because it comes after the sleep homeostat, but is not an abbreviation for the homeostat. 

2. The abstract discusses dFB, but the introduction describes R23E10 neurons, and then the results describe R23E10 ad projecting onto the dFB. Clearly laying out the circuit in the abstract/into would be helpful.

3. I find this sentence awkward. It would seem proper to thank them in the acknowledgments rather than describing the request in the results section. 'To expedite the identification of such lines, we asked Dr. Gerry Rubin and Dr. Arnim Jenett, for assistance [49]'

4. Check to make sure the Epac transgene is consistently written throughout the manuscript text and figures

5. In discussion, suggest changing the phrase 'historical context' to something more specific.

---

## [Decision Letter · Decision Letter 2]

31 May 2022

Dear Dr Shaw,

Thank you for your patience while we considered your revised manuscript "Sleep promoting neurons remodel their response properties to calibrate sleep drive with environmental demands" for publication as a Research Article at PLOS Biology. Your revised study has been evaluated by the PLOS Biology editors, the Academic Editor and the original reviewers.

The reviews are appended below. As you will see, both Reviewers 2 and 3 are satisfied by the revision, however Reviewer 1 has noted that one of the key conclusions is still not adequately supported. While Reviewer 1 does acknowledge that the conclusions are currently written with the appropriate caveats, after discussion with the Academic Editor, we think it would be important to strengthen this conclusion with new data to more tightly link behavior and neurophysiology for a key point in the study, especially as the reagents and methodology are available.

Although we usually would not invite a second round of experimental revision, in this case we would like to give you the opportunity to address Reviewer 1’s comments and provide the requested Dop1R1 RNAi experiment. As you address the remaining reviewer comments, we also ask that you address the following editorial requests: 

1 - Data request - Thank you for providing the underlying data for your figures. I could not find a corresponding excel file for a couple of figures or panels within these excel files (for example, I did not see the underlying data for Figure S2C or Figure S9). Can you please ensure that the supplementary data files contain underlying data for all relevant figures? 

Please also ensure that figure legends in your manuscript include information on where the underlying data can be found, and ensure your supplemental data files have legends.

For more information on the PLOS Data policy, please see http://journals.plos.org/plosbiology/s/data-availability and this editorial: http://dx.doi.org/10.1371/journal.pbio.1001797

2. Species in abstract? - Please note that per journal policy, the model system/species studied should be clearly stated in the abstract of your manuscript. 

3. Data not shown - Please note that per journal policy, we do not allow the mention of "data not shown", "personal communication", "manuscript in preparation" or other references to data that is not publicly available or contained within this manuscript. Please either remove mention of these data or provide figures presenting the results and the data underlying the figure(s).

4. Blurb - Please provide a blurb which (if accepted) will be included in our weekly and monthly Electronic Table of Contents, sent out to readers of PLOS Biology, and may be used to promote your article in social media. The blurb should be about 30-40 words long and is subject to editorial changes. It should, without exaggeration, entice people to read your manuscript. It should not be redundant with the title and should not contain acronyms or abbreviations.

Given the extent of revision needed, we cannot make a decision about publication until we have seen the revised manuscript and your response to the reviewers' comments. Your revised manuscript may also be sent for further evaluation by a subset of the reviewers.

**IMPORTANT - SUBMITTING YOUR REVISION**

*Re-submission Checklist*

*Published Peer Review*

Sincerely,

Luke

Lucas Smith, Ph.D.

Associate Editor

PLOS Biology

lsmith@plos.org

REVIEWS:

Reviewer #1: The authors have mostly addressed my and the other reviewers' comments, but I would like to clarify one point. In response to my suggestion that "showing that RNAi against Dop1R1 abrogates the effect would be the most direct demonstration" they responded: 

"An increase in sleep is observed after both Time Restricted Feeding (Figure 5) and a night of starvation (Figure 6) when the Dop1R1 is knocked down in R23E10 and R55B01 neurons. That is, both starvation paradigms yield similar outcomes.

R23E10 neurons express Dop1R2 under baseline conditions. As a consequence, knocking down the Dop1R1 will not prevent R23E10 neurons from responding to Dopamine. It is only by knocking down the

Dop1R2 and demonstrating that R23E10 neurons no longer respond to Dopamine under baseline conditions that we can evaluate the role of Dop1R1 after starvation."

My point was that they show in Figure 7D that the dopamine response comes back in recovery, even when Dop1R2 is absent. To complete the logical circuit, though, they should show also now that Dop1R1 RNAi inhibits the dopamine response in this recovery phase. Otherwise, they cannot formally rule out the possibility that this dopamine signal isn't derived from another source. I note they do concede in the text that, "While we cannot exclude a role of other Dopamine receptors (e.g. Dopamine/Ecdysteroid receptor) for the observed changes, when viewed with the sleep experiments shown above (Figures 5 and 6), these data suggest that during recovery from starvation the constellation of Dopamine receptors in R23E10 and R55B01 changes and most likely includes the recruitment of Dop1R1", which is perhaps good enough. However, I believe Reviewer 2 is making a similar set of points in point 4, "it is better to include the live imaging data of Dop1R1 knockdown before and after TRF"-- highlighting the importance of this point to support some of the most interesting findings of this paper. 

Reviewer #2: I think the authors replied to all of my comments.

Reviewer #3: The authors have addressed all my outstanding concerns, and the manuscript has been significantly strengthened.

---

## [Editor Report · Decision Letter 3]

2 Aug 2022

Dear Paul,

Thank you for submitting revised manuscript "Sleep promoting neurons remodel their response properties to calibrate sleep drive with environmental demands" for publication as a Research Article at PLOS Biology. This revised version of your manuscript has been evaluated by the PLOS Biology editors and the Academic Editor.

We were persuaded by the arguments you made in your response to the request from Reviewer 1 and overall, we are largely satisfied by the changes made in this revision. However, there is one important lingering issue which will need to be addressed before we can editorially accept your manuscript for publication. In our last decision letter we noted several instances where conclusions were supported by the statement "data not shown", which is against journal policy. In the revised manuscript, it seems that the statement "data not shown" was removed, but the data necessary to support those conclusions was not added. After discussion with the Academic Editor, we think that it would be important for your to add figures/reference existing data, as appropriate, to support those claims.

For example, please provide additional figures to support the following conclusions:

1) “Similar results were obtained when expressing the sodium bacterial channel, UAS-NaChBac”

2) “The response properties of R23E10 neurons to Dopamine were similar in 6-8 day old flies and 30-38 day old flies. In contrast to age, we did not observe any changes in the response properties of R23E10 neurons in male and female flies despite large sexual dimorphisms in sleep behavior [52, 54]”.

3) “Furthermore, no changes in responses to Dopamine were found when training was followed by 4 h of sleep deprivation”

4) “R23E10>UAS-Dop1R2RNAi flies displayed an increase in sleep under baseline conditions consistent with previous reports”

**Important: as you add these additional figures, please make sure to update your S1_Data file to include the underlying data for any new graphs.

We expect to receive your revised manuscript within two weeks, but do let us know if you need more time to address this request. 

*Published Peer Review History*

*Press*

Sincerely,

Luke

Lucas Smith, Ph.D.

Associate Editor,

lsmith@plos.org,

PLOS Biology

---

## [Editor Report · Decision Letter 4]

16 Aug 2022

Dear Paul,

Thank you for the submission of your revised Research Article "Sleep promoting neurons remodel their response properties to calibrate sleep drive with environmental demands" for publication in PLOS Biology, and for addressing our last editorial requests. On behalf of my colleagues and the Academic Editor, Richard Benton, I am pleased to say that we can in principle accept your manuscript for publication, provided you address any remaining formatting and reporting issues. These will be detailed in an email you should receive within 2-3 business days from our colleagues in the journal operations team; no action is required from you until then. Please note that we will not be able to formally accept your manuscript and schedule it for publication until you have completed any requested changes.

PRESS

Sincerely, 

Luke

Lucas Smith, Ph.D., Ph.D.

Associate Editor

PLOS Biology

lsmith@plos.org